JCB  Journal of Cell Biology

# Multifaceted modes of γ-tubulin complex recruitment and microtubule nucleation at mitotic centrosomes

Zihan Zhu[1] , Isabelle Becam[2] , Corinne A. Tovey[2] , Abir Elfarkouchi[2] , Eugenie C. Yen[1] , Fred Bernard[2] , Antoine Guichet[2] , and Paul T. Conduit[1,2]

Microtubule nucleation is mediated by γ-tubulin ring complexes (γ-TuRCs). In most eukaryotes, a GCP4/5/4/6 "core" complex promotes γ-tubulin small complex (γ-TuSC) association to generate cytosolic γ-TuRCs. Unlike γ-TuSCs, however, this core complex is non-essential in various species and absent from budding yeasts. In *Drosophila*, Spindle defective-2 (Spd-2) and Centrosomin (Cnn) redundantly recruit γ-tubulin complexes to mitotic centrosomes. Here, we show that Spd-2 recruits γ-TuRCs formed via the GCP4/5/4/6 core, but Cnn can recruit γ-TuSCs directly via its well-conserved CM1 domain, similar to its homologs in budding yeast. When centrosomes fail to recruit γ-tubulin complexes, they still nucleate microtubules via the TOG domain protein Mini-spindles (Msps), but these microtubules have different dynamic properties. Our data, therefore, help explain the dispensability of the GCP4/5/4/6 core and highlight the robustness of centrosomes as microtubule organizing centers. They also suggest that the dynamic properties of microtubules are influenced by how they are nucleated.

## Introduction

During cell division, centrosomes act as major microtubule organizing centers (MTOCs) to nucleate and organize microtubules that contribute to mitotic spindle formation (Conduit et al., 2015). Centrosomes comprise a pair of centrioles that recruit and are surrounded by the pericentriolar material (PCM). The PCM is a large collection of proteins and is the predominant site of microtubule nucleation and organization during mitosis. On entry into mitosis, centrosomes expand their PCM in a process called centrosome maturation (Khodjakov and Rieder, 1999; Piehl et al., 2004). This is particularly dramatic in *Drosophila* cells because interphase centrosomes have very little PCM and do not organize microtubules, while mitotic centrosomes have relatively large amounts of PCM and robustly organize microtubules (Rogers et al., 2008). This makes *Drosophila* centrosomes ideal for the study of mitotic PCM assembly and microtubule nucleation.

γ-Tubulin ring complexes (γ-TuRCs) are important PCM clients because they template the nascent assembly of microtubules (microtubule nucleation; Tovey and Conduit, 2018; Farache et al., 2018; Kollman et al., 2011). Along with actin and Mozart proteins, they comprise a single-turn helical arrangement of 14 laterally associated "spokes," each made from a γ-tubulin complex protein (GCP or "Grip" protein in *Drosophila*) and a γ-tubulin molecule. The essential subunits of γ-TuRCs are

2-spoke γ-tubulin small complexes (γ-TuSCs), made from GCP2, GCP3, and two γ-tubulins. In budding yeast, γ-TuSCs are stimulated to assemble into helical structures when bound by the conserved "Centrosomin motif 1" (CM1) domain found within the yeast spindle pole body (SPB; centrosome equivalent) proteins, Spd110 and Spc72 (Stu2, a TOG domain protein, is also required in the case of Spc72; Kollman et al., 2010; Gunzelmann et al., 2018). The "CM1 motif" within Spc110's CM1 domain binds across adjacent γ-TuSCs, which presumably promotes the oligomerization process at the SPB (Brilot et al., 2021; Lyon et al., 2016; Kollman et al., 2010; Lin et al., 2014). In most eukaryotes, however, γ-TuSCs are stimulated to assemble into γ-TuRCs within the cytosol via a 4-spoke GCP4/5/4/6 core complex that seeds ring assembly and is absent from budding yeast (Haren et al., 2020; Würtz et al., 2022). Indeed, the depletion of GCP4, GCP5, or GCP6 strongly inhibits cytosolic γ-TuRC assembly in humans, *Xenopus*, *Drosophila*, *Aspergillus*, and fission yeast cells (Cota et al., 2017; Vogt et al., 2006; Vérollet et al., 2006; Xiong and Oakley, 2009; Zhang et al., 2000). Intriguingly, however, these γ-TuRC-specific proteins are not essential in *Drosophila*, *Aspergillus*, or fission yeast (Xiong and Oakley, 2009; Vogt et al., 2006; Anders et al., 2006; Vérollet et al., 2006). Consistent with this, γ-TuSCs can be recruited to *Drosophila* S2 cell centrosomes after the depletion of GCP4/5/4/6 core complex components

[1]Department of Zoology, University of Cambridge, Cambridge, UK; [2]Université Paris Cité, CNRS, Institut Jacques Monod, Paris, France.

Correspondence to Paul T. Conduit: paul.conduit@ijm.fr.

(Vérollet et al., 2006) and are recruited independently of the GCP4/5/4/6 core complex to the outer SPB plaque in *Aspergillus* (Gao et al., 2019). Nevertheless, how γ-TuSCs are recruited to centrosomes in the absence of the GCP4/5/4/6 core remains unclear.

The predominant view of γ-TuRC recruitment involves the binding of large coiled-coil "tethering proteins" whose experimental depletion leads to measurable reductions in γ-tubulin levels at centrosomes. One of these proteins, NEDD1/Grip71, associates with preformed γ-TuRCs in the cytosol and subsequently docks γ-TuRCs to the centrosomes (Lüders et al., 2006; Haren et al., 2006, 2009; Zhang et al., 2009; Gomez-Ferreria et al., 2012a, 2012b). All other tethering proteins do not associate with cytosolic γ-TuRCs but instead localize to centrosomes and appear to "dock" incoming γ-TuRCs. These include the Pericentrin family of proteins, CM1 domain-containing proteins (e.g., human CDK5RAP2, *Drosophila* Centrosomin [Cnn], fission yeast Mto1, and budding yeast Spc110 and Spc72), and the Spd-2 family of proteins (CEP192 in humans; Zimmerman et al., 2004; Gomez-Ferreria et al., 2007; Haren et al., 2009; Fong et al., 2008; Sawin et al., 2004; Zhang and Megraw, 2007; Conduit et al., 2014; Dobbelaere et al., 2008; Lee and Rhee, 2011). It is difficult to determine the individual role of these proteins in γ-TuRC recruitment as they act redundantly and depend on each other for their proper localization within the PCM. We previously showed that γ-tubulin partially accumulated at mitotic centrosomes in the absence of either Cnn or Spd-2, but failed to accumulate when both proteins were removed, with centrosomes also failing to accumulate other PCM proteins and nucleate microtubules (Conduit et al., 2014). This data showed that Cnn and Spd-2 can independently recruit γ-tubulin-containing complexes (hereafter γ-tubulin complexes), but it remained unclear how.

Cnn contains the highly conserved CM1 domain (Sawin et al., 2004), which binds directly to γ-tubulin complexes in yeast and humans, respectively, (Brilot et al., 2021; Wieczorek et al., 2019; Choi et al., 2010) and has been implicated in the recruitment of γ-tubulin complexes to centrosomes in different systems (Zhang and Megraw, 2007; Lyon et al., 2016; Samejima et al., 2008; Choi et al., 2010; Muroyama et al., 2016; Fong et al., 2008). However, whether the CM1 domain is essential for Cnn to recruit γ-tubulin complexes remains unclear as the effect of removing the CM1 domain has only been tested in the presence of Spd-2 (Zhang and Megraw, 2007). In contrast to Cnn, Spd-2 does not contain a CM1 domain and so how it recruits γ-tubulin complexes remains to be established.

In this study, we investigated how γ-tubulin complexes are recruited to mitotic centrosomes by Cnn and Spd-2. We used classical *Drosophila* genetics to combine specific mutant alleles or RNAi constructs and examined γ-tubulin accumulation at centrosomes in larval brain cells. We show that Cnn allows the centrosomal accumulation of γ-tubulin in the absence of the GPC4/5/4/6 core and Grip71 and that this is dependent on its CM1 domain. Mutations in the CM1 domain also abolish Cnn's ability to associate with γ-tubulin in immunoprecipitation assays. This suggests that Cnn's CM1 domain can bind and recruit γ-TuSCs to centrosomes, similar to the CM1 domains in budding

yeast Spc110 and Spc72. In contrast, we find that Spd-2 does not support the centrosomal accumulation of γ-tubulin in the absence of the GPC4/5/4/6 core and Grip71, suggesting that Spd-2 can only recruit γ-TuRCs that have preformed in the cytosol. By selectively abolishing γ-tubulin complex recruitment, we show that mitotic centrosomes can nucleate microtubules independently of γ-tubulin complexes and that this depends on the TOG domain protein Mini-spindles (Msps), consistent with the conserved ability of TOG domain proteins to promote microtubule nucleation in vitro. Moreover, the microtubules nucleated in the absence of γ-TuRCs are more cold-stable than those nucleated in the presence of γ-TuRCs, suggesting that the dynamic properties of microtubules depend in part on how the microtubules were nucleated.

## Results

### Cnn recruits γ-tubulin complexes via its CM1 domain independently of Grip71 and the GCP4/5/4/6 core

We first explored how Cnn recruits γ-tubulin complexes to mitotic centrosomes by comparing the levels of centrosomal γ-tubulin at interphase and mitotic centrosomes in larval brain cells from flies depleted of Spd-2 and different γ-tubulin complex proteins. Typically, interphase centrosomes have only ~5–20% of the γ-tubulin levels found at mitotic centrosomes, and this residual γ-tubulin is closely associated with the centrioles and is non-functional with respect to microtubule nucleation (Conduit et al., 2014). An increase in γ-tubulin signal between interphase and mitotic centrosomes indicates that γ-tubulin complexes have been recruited to the expanding mitotic PCM, at least to some degree. Similar to our previous results (Conduit et al., 2014), we found that γ-tubulin was recruited to mitotic centrosomes in *spd-2* null mutant brains with an average level of ~77% compared with wild-type brains (Fig. 1, A and B). We know that this recruitment of γ-tubulin is entirely dependent on Cnn because centrosomes in *cnn,spd2* double mutants fail entirely to recruit γ-tubulin during mitosis (Conduit et al., 2014). In contrast, combining *spd-2* null mutant alleles with null or severe depletion mutant alleles, or RNAi alleles, for Grip71 and the GCP4/5/4/6 core components did not prevent γ-tubulin accumulation at mitotic centrosomes. In fact, the centrosomes in *spd-2,grip71,grip75^{GCP4},grip128^{GCP5}*-RNAi,*grip163^{GCP6}* mutant cells had ~66% of the γ-tubulin levels found at wild-type centrosomes, only slightly lower than the ~77% in *spd-2* mutants alone (Fig. 1, A and B). Thus, the recruitment of γ-tubulin to mitotic centrosomes that occurs in the absence of Spd-2, i.e., that depends upon Cnn, does not appear to require Grip71 or the GCP4/5/4/6 core.

While we cannot rule out that residual amounts of GCP4/5/4/6 core components in *spd-2,grip71,grip75^{GCP4},grip128^{GCP5}*-RNAi,*grip163^{GCP6}* mutant cells may support a certain level of γ-TuSC oligomerization in the cytosol, we favor the conclusion that Cnn can recruit γ-TuSCs directly to centrosomes in the absence of the GCP4/5/4/6 core for several reasons: first, the alleles used for *grip71* and *grip75^{GCP4}* are null mutants and the allele for *grip163^{GCP6}* is a severe depletion allele (see Materials and methods), and even individual mutations in, or RNAi-

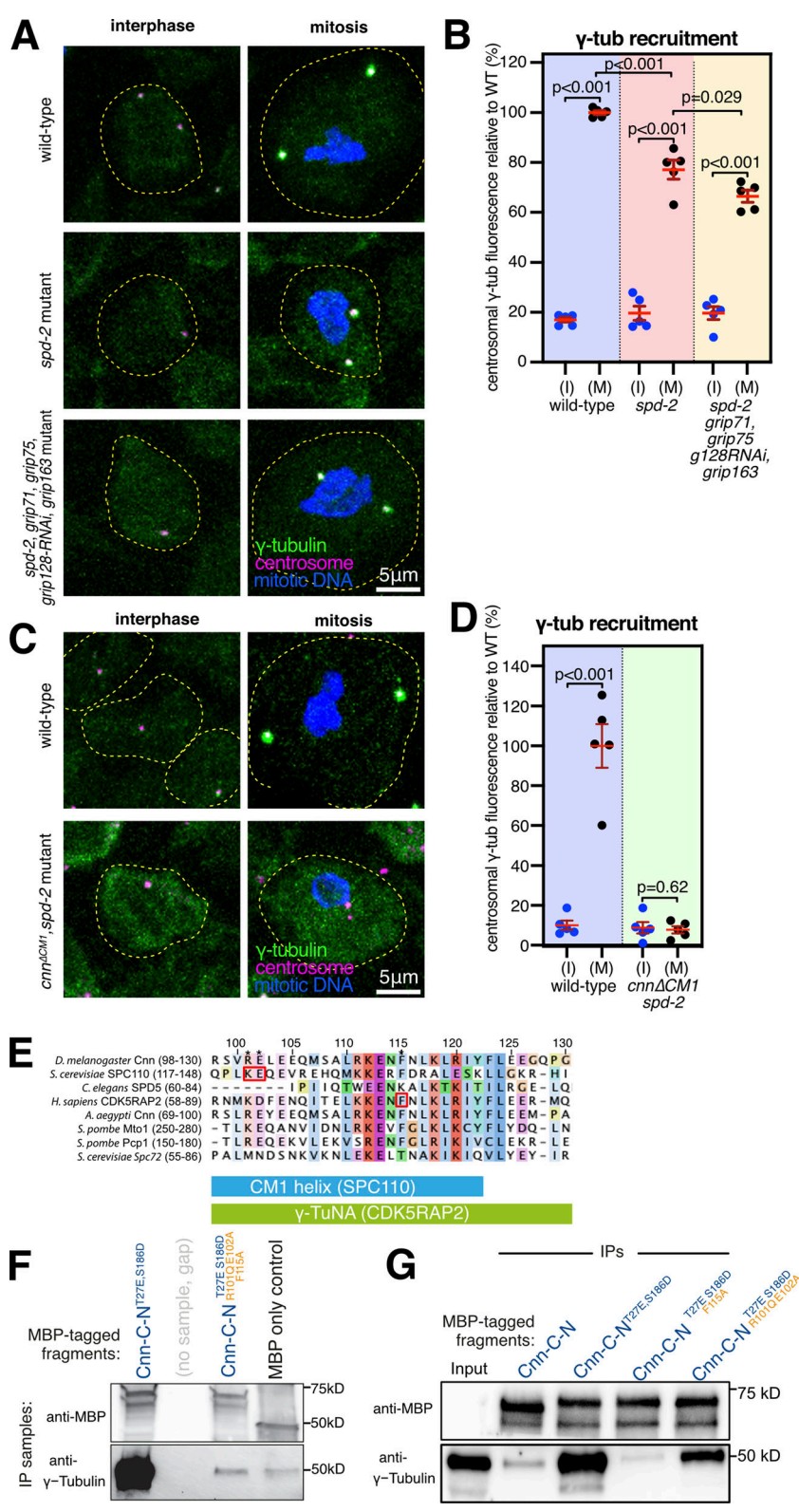

Figure 1. **Cnn recruits γ-tubulin complexes via its CM1 domain and independently of Grip71 and the GCP4/5/4/6 core. (A)** Fluorescence images of either interphase or mitotic *Drosophila* brain cells from either wild-type, *spd-2* mutant, or *spd-2, grip71,grip75*GCP4*,grip128*GCP5-RNAi,*grip163*GCP6 mutant third instar larval brains immunostained for γ-tubulin (green), mitotic DNA (blue), and Asl (centrioles, magenta). Both mutants carry the mutant *spd-2* alleles to reveal the Cnn pathway of recruitment. The scale bar is 5 µm and applies to all images. **(B)** Graph showing average centrosomal fluorescence intensities of γ-tubulin (relative to wild type) of interphase (blue dots) and mitotic (black dots) centrosomes from different genotypes (as indicated below). Each datapoint represents the average centrosome value from one brain. N = 5 for each condition. Mean and SEM are indicated. A one-way ANOVA with a Sidak's multiple comparisons test was used to make the comparisons indicated by P values in the graph. Note that there is only a small reduction in mitotic centrosomal γ-tubulin levels in *spd-2* mutants and in *spd-2,grip71,grip75*GCP4*,grip128*GCP5-RNAi,*grip163*GCP6 mutants, showing that Cnn can still efficiently recruit γ-tubulin complexes to mitotic centrosomes when only γ-TuSCs are present. **(C)** Fluorescence images of either interphase or mitotic *Drosophila* brain cells from either wild-type or *cnn*ΔCM1*,spd-2* mutant third instar larval brains immunostained for γ-tubulin (green), mitotic DNA (blue), and Asl (centrioles, magenta). The scale bar is 5 µm and applies to all images. **(D)** The graph is in the same format as in B, revealing no significant increase of centrosomal γ-tubulin signal from interphase to mitosis in *cnn*ΔCM1*; spd-2* mutant cells, showing that Cnn requires its CM1 domain to recruit γ-tubulin complexes to centrosomes. Two-sided paired *t* tests were used to compare mean values of interphase and mitotic centrosomes within each genotype. **(E)** Multiprotein sequence alignment of part of the CM1 domain containing the key binding residues (indicated by red boxes) in budding yeast and humans that we mutated in *Drosophila*. **(F and G)** Western blots probed for MBP and γ-tubulin showing the results of IP experiments from wild-type embryo extracts using bacterially purified MBP-tagged N-terminal (aa1–255) Cnn fragments containing point mutations to relieve Cnn-C autoinhibition (T27E and S186D; Tovey et al., 2021) and to perturb the CM1 domain's ability to bind γ-TuRCs (R101Q, E102A, and F115A). Source data are available for this figure: SourceData F1.

directed depletion of, Grip75GCP4, Grip128GCP5 or Grip163GCP6 are sufficient to strongly reduce the presence cytosolic γ-TuRCs (Vogt et al., 2006; Vérollet et al., 2006). Second, *spd-2, grip71,grip75*GCP4*,grip128*GCP5-RNAi,*grip163*GCP6 mutant cells are depleted for all structural γ-TuRC components except for γ-TuSCs and Actin (note that Mozart1 [Mzt1] is not expressed in

larval brain cells [Tovey et al., 2018] and that Mzt2 has not been identified in flies). In human and *Xenopus* γ-TuRCs, Actin supports γ-TuRC assembly via interactions with a GCP6-N-term-Mzt1 module (Liu et al., 2019; Wieczorek et al., 2019, 2020; Zimmermann et al., 2020; Consolati et al., 2020), and so Actin alone is unlikely to facilitate assembly of γ-TuSCs into higher

order structures. Third, our data agree with the observation that near complete depletion of Grip71, Grip75GCP4, Grip128 GCP5, and Grip163GCP6 from S2 cells does not prevent γ-tubulin recruitment to centrosomes (Vérollet et al., 2006). Fourth, given the strength of mutant alleles used, one would have expected a much larger decrease in centrosomal γ-tubulin levels in *spd-2,grip71,grip75GCP4,grip128GCP5*-RNAi,*grip163GCP6* mutant cells if Cnn were not able to recruit γ-TuSCs directly to centrosomes. Thus, Cnn appears to recruit γ-TuSCs to centrosomes without a requirement for them to first assemble into higher-order complexes.

The recruitment of γ-TuSCs to centrosomes by Cnn appears to reflect the natural situation in budding yeast, where homologs of the GCP4/5/4/6 core, Grip71 and Mzt1, are absent and where γ-TuSCs are recruited to the SPB by direct binding of Spc110 and Spc72's CM1 domain. We, therefore, reasoned that Cnn's recruitment of γ-TuSCs may also be mediated by its CM1 domain. A previous study, however, had shown that replacing the endogenous *cnn* gene with an ectopically expressed UAS-GFP-Cnn construct lacking the CM1 domain led to a reduction, but not elimination, of γ-tubulin at centrosomes in syncytial embryos (Zhang and Megraw, 2007). Along with the potential effects caused by Cnn over-expression, we now know that Spd-2 can recruit γ-tubulin complexes independently of Cnn (Conduit et al., 2014), making it hard to evaluate the true effect of deleting the CM1 domain without also removing Spd-2. We, therefore, deleted the CM1 domain (amino acids 98–167, inclusive) from the endogenous *cnn* gene (Fig. S1, A–C; see Materials and methods) and combined this mutant allele with the *spd-2* null mutant allele. Note that the Cnn$^{\Delta CM1}$ protein was produced at similar levels to wild-type Cnn (Fig. S1 D). We found that γ-tubulin no longer accumulated at mitotic centrosomes in these *cnn$^{\Delta CM1}$,spd-2* mutant cells (Fig. 1, C and D), showing definitively that Cnn's CM1 domain is essential for Cnn to recruit γ-tubulin complexes to mitotic centrosomes.

We also tested whether the CM1 domain was required for Cnn to associate with γ-tubulin complexes by comparing the ability of bacterially purified MBP-tagged Cnn fragments to coimmunoprecipitate γ-tubulin from wild-type cytosolic embryo extracts. We recently showed that Cnn's centrosomal isoform (Cnn-C) is autoinhibited from binding cytosolic γ-tubulin complexes by an extreme N-terminal "CM1 auto-inhibition" (CAI) domain, but that this autoinhibition can be relieved by introducing T27E and S186D phospho-mimetic mutations (Tovey et al., 2021). These mutations were therefore included in the fragments to enable Cnn binding in "control" conditions (Cnn-C-N$^{T27E,S186D}$). To identify mutations predicted to perturb CM1 binding, we used cross-species protein sequence alignments and identified F115, R101, and E102 as equivalent to residues important for γ-tubulin complex binding in humans (F75; Choi et al., 2010) and budding yeast (K120 and E121; Lin et al., 2014; Gunzelmann et al., 2018; Fig. 1 E). We mutated these residues in the Cnn-C-N$^{T27E,S186D}$ fragments (R101Q, E102A, and F115A) to mimic the mutations previously used in yeast and human experiments. Introducing all three mutations, or only F115A, abolished the ability of the Cnn fragments to coimmunoprecipitate γ-tubulin, while introducing E102A and F115A reduced but

did not abolish coimmunoprecipitation (Fig. 1, F and G). Thus, F115 within Cnn's CM1 domain is essential for Cnn binding to γ-TuRCs, as is true of the equivalent F75 residue in human CDK5RAP2.

We conclude that the CM1 domain is essential for Cnn binding to γ-TuRCs. Moreover, taken together, our data strongly indicate that, similar to Spc110 and Spc72 in budding yeast, Cnn can bind and recruit γ-TuSCs to centrosomes directly from the cytosol without the need for them to preform into γ-TuRCs in the cytosol.

## Spd-2 predominantly recruits preformed γ-TuRCs from the cytosol

To explore how Spd-2 recruits γ-tubulin complexes even though (unlike Cnn) it lacks a CM1 domain, we compared the levels of centrosomal γ-tubulin at interphase and mitotic centrosomes in larval brain cells from flies lacking Cnn and different γ-tubulin complex proteins. In *cnn* mutants alone, Spd-2 levels at mitotic centrosomes are reduced to ∼60% (Conduit et al., 2014), and this Cnn-independent pool of Spd-2 recruits γ-tubulin to on average ∼22–23% of wild-type levels (Fig. 2, A and B; Conduit et al., 2014). We were therefore testing which γ-TuRC components, when removed in addition to Cnn, reduced the mitotic centrosomal level of γ-tubulin further, such that there was no significant accumulation of γ-tubulin above interphase levels. Of note, the Cnn-dependent pool of Spd-2 also recruits some γ-tubulin complexes because γ-tubulin levels were slightly higher at centrosomes in Cnn$^{\Delta CM1}$ mutant cells compared with *cnn* null mutant cells (Fig. S1, E and F). The absence of a large increase may be because deleting the CM1 domain appears to affect the ability of Cnn to form a robust scaffold and therefore maintain Spd-2 in the PCM, as we noticed that the γ-tubulin signals were often offset from the centriole signal (Fig. S1 E), as is the case in *cnn* mutant cells (Fig. 2 A; Fig. S1 E; Lucas and Raff, 2007).

We predicted that Spd-2 recruits γ-tubulin complexes via Grip71 because the human homolog of Spd-2, CEP192, associates with the human homolog of Grip71, NEDD1 (Gomez-Ferreria et al., 2012a). We found, however, that γ-tubulin could still accumulate at mitotic centrosomes relatively well in *cnn,grip71* mutant cells, being only slightly lower than the γ-tubulin levels in *cnn* mutant cells (Fig. 2, A and B). Thus, Spd-2 relies only partly on Grip71 to recruit γ-tubulin complexes. There was a stronger reduction, however, when we removed Cnn and members of the GCP4/5/4/6 core, Grip75GCP4, and Grip163GCP6, either individually or in combination (Fig. 2, A and B). Given that the GCP4/5/4/6 core is required for the assembly of γ-TuRCs within the cytosol, this result suggests that Spd-2 (unlike Cnn) predominantly recruits preformed γ-TuRCs rather than γ-TuSCs. We found, however, that γ-tubulin accumulation at mitotic centrosomes was only abolished after the additional removal of Grip71 i.e., in *cnn,grip71,grip163GCP6* mutant cells (Fig. 2, A and B), a phenotype that was not due to a failure of Spd-2 to accumulate at mitotic centrosomes (Fig. 2, C and D). Thus, Spd-2 appears to recruit a very small amount of γ-TuSCs (which may, or may not, be present as larger assemblies due to an association with Grip128-γ-tubulin) via Grip71 (i.e., the

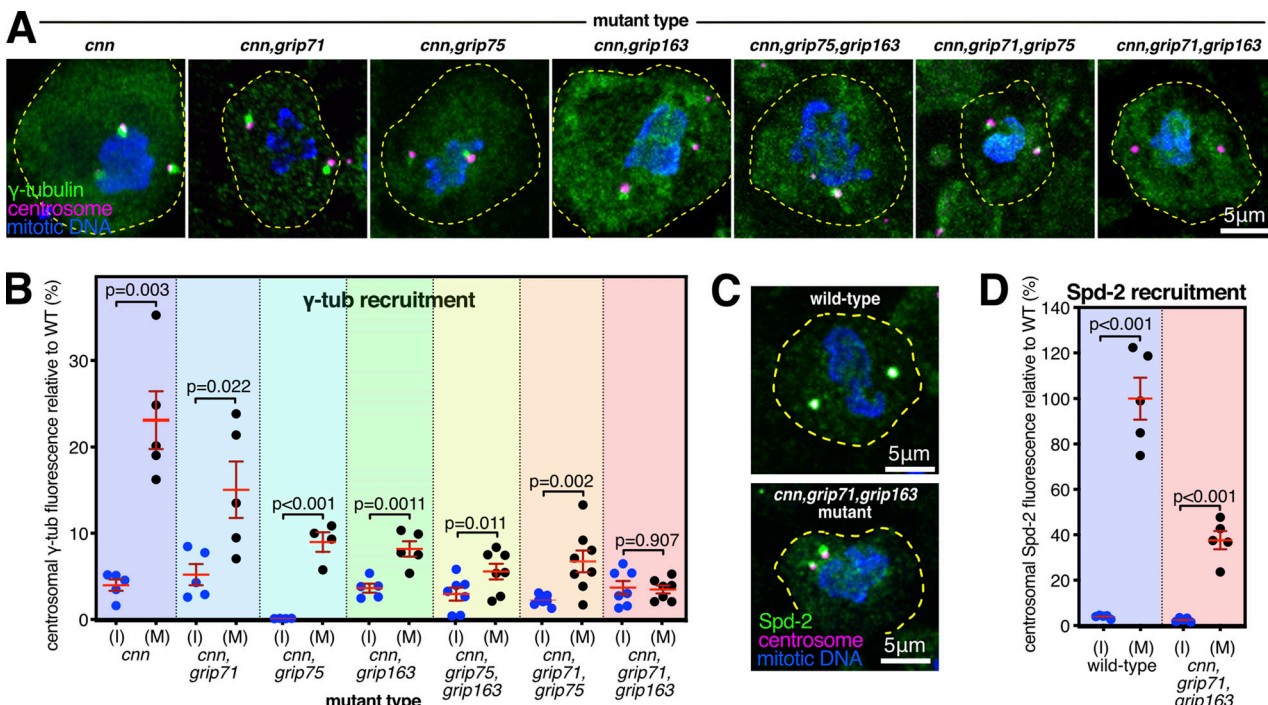

**Figure 2. Recruitment of γ-tubulin complexes by Spd-2 is heavily dependent on the GCP4/5/4/6 core. (A)** Fluorescence images of mitotic *Drosophila* brain cells from various mutant third instar larvae immunostained for γ-tubulin (green), mitotic DNA (blue), and Asl (centrioles, magenta). All mutants carry the mutant *cnn* allele to reveal the Spd-2 pathway of recruitment, along with mutant alleles for different combinations of γ-TuRC genes. Scale bar is 5 μm and applies to all images. **(B)** Graph showing average centrosomal fluorescence intensities of γ-tubulin (relative to wild-type) of interphase (blue dots) and mitotic (black dots) centrosomes from different genotypes (as indicated below). Each datapoint represents the average centrosome value from one brain. *N* numbers are the same interphase and mitosis of each condition: five for *cnn,cnn,grip71*, and *cnn,grip163*; four for *cnn,grip75*; seven for *cnn,grip71,grip163* and *cnn,grip75,grip163*; and eight for *cnn,grip71,grip75*. Mean and SEM are indicated. Two-sided paired *t* tests were used to compare mean values of interphase and mitotic centrosomes within each genotype. Note that γ-tubulin accumulation at mitotic centrosomes is severely perturbed in the absence of the GCP4/5/4/6 core components Grip75[GCP4] and Grip163[GCP6], but is abolished only in the absence of Grip71 and Grip163[GCP6]. **(C)** Fluorescence images of mitotic *Drosophila* brain cells from wild-type or *cnn,grip71,grip163* mutant third instar larvae immunostained for Spd-2 (green), mitotic DNA (blue), and Asl (centrioles, magenta). Scale bar is 5 μm and applies to both images. **(D)** Graph is in the same format as in B using two-sided paired *t*-tests to show a significant increase of centrosomal Spd-2 signal from interphase to mitosis in *cnn,grip71,grip163* mutant cells, showing that the inability of these centrosomes to recruit γ-tubulin (A and B) is not due to an absence of Spd-2. *N* = 5 for all conditions.

recruitment that occurs in cnn,grip75[GCP4],grip163 [GCP6] cells), but its recruitment of γ-tubulin complexes relies predominantly on the GCP4/5/4/6 core.

Intriguingly, γ-tubulin could still accumulate at mitotic centrosomes to some degree in cells lacking Cnn, Grip71, and Grip75[GCP4], showing that removal of Grip75[GCP4] does not perfectly phenocopy the removal of Grip163[GCP6], and therefore suggesting that Grip163[GCP6] may still be able to promote at least partial γ-TuRC assembly in the absence of Grip75[GCP4]. This is consistent with observations in human cells, where GCP6 depletion has a greater effect on cytosolic γ-TuRC assembly than GCP4 depletion (Cota et al., 2017). Alternatively, Spd-2 may interact with Grip163[GCP6] and so be able to recruit its associated γ-tubulin independent of Grip75[GCP4].

In summary, Spd-2's recruitment of γ-TuRCs relies strongly on the presence of the GCP4/5/4/6 core, and therefore on γ-TuRC assembly within the cytosol, but the additional removal of Grip71 is required to entirely prevent accumulation of γ-tubulin at mitotic centrosomes. In contrast, Cnn's conserved CM1 domain can mediate the recruitment of γ-TuSCs directly from the cytosol. The requirement of Spd-2 for the GCP4/5/4/6

core aligns with the absence of Spd-2 and GCP4/5/4/6 core component homologs in lower eukaryotes. In addition, the ability of Cnn to recruit γ-TuSCs may explain why the GCP4/5/4/6 core is not essential in several species studied so far, particularly if all CM1 domain proteins are able to stimulate γ-TuSC assembly into ring-like structures, as is the case for yeast Spc110 and Spc72.

## Centrosomes lacking γ-tubulin can still nucleate and organize microtubules

In the course of examining *cnn,grip71,grip163* mutants, we observed that their mitotic centrosomes, which fail to accumulate γ-tubulin but still accumulate Spd-2, were still associated with microtubules during prophase and localized to spindle poles during mitosis (Fig. 3 A). This is in contrast to centrosomes in *cnn,spd-2* mutant cells, which lack PCM entirely, fail to nucleate or organize microtubules, and do not associate with spindle poles (Conduit et al., 2014). Thus, mitotic centrosomes can organize microtubules independently of γ-TuRCs so long as the PCM can at least partially assemble. To test whether these microtubules are actually nucleated at centrosomes (rather than

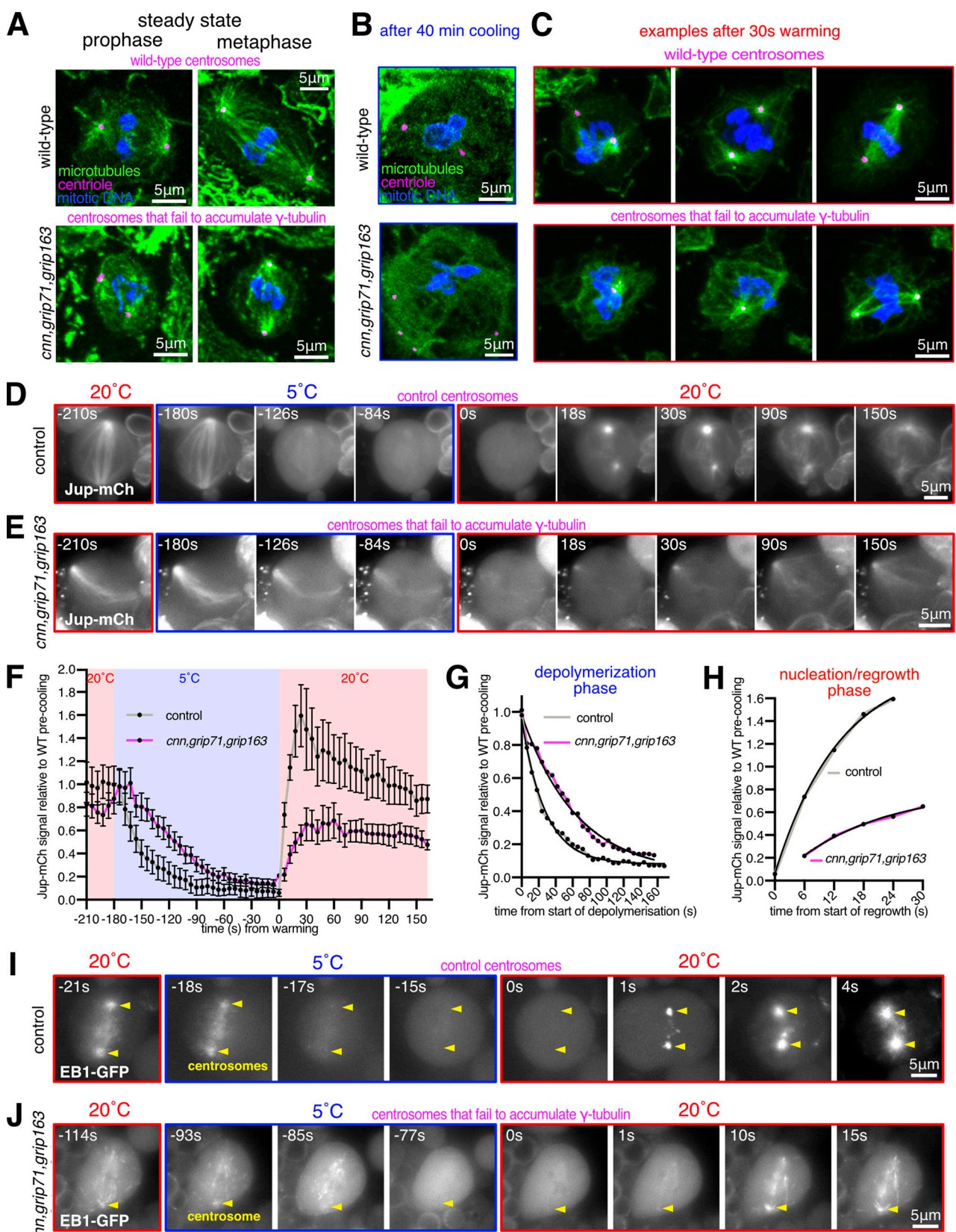

Figure 3. **Mitotic centrosomes that fail to accumulate γ-tubulin can still nucleate microtubules. (A–C)** Fluorescence images of mitotic *Drosophila* brain cells from either wild-type or *cnn,grip71,grip163* mutant third instar larval brains, either at steady state (A), after 40 min of cooling on ice (B), or after 30 s of warming (post cooling) to room temperature (C) immunostained for alpha-tubulin (microtubules, green), mitotic DNA (blue), and Asl (centrioles, magenta). Note how centrosomes in both wild-type and *cnn,grip71,grip163* mutant cells are associated with microtubules both at steady state and after 30 s warming.

Note that some cells lacking Cnn have abnormal numbers of centrosomes due to centrosome segregation problems during cell division (Conduit et al., 2010). **(D–F)** Fluorescent images (D and E) and graph (F) documenting the behavior of the microtubule marker Jupiter-mCherry within living *Drosophila* control (D) or *cnn,grip71,grip163* mutant (E) third instar larval brain cells as they were cooled to 5°C for ~3 min and then rapidly warmed to 20°C. Time in seconds relative to the initiation of warming (0 s) is indicated. Note that the GFP-PACT signal used to locate centrosomes is not displayed. The graph in F plots the mean and SEM centrosomal signal (after subtraction of cytosolic signal) of 12 and 10 centrosomes from 7 and 10 control and *cnn,grip71,grip163* mutant cells, respectively. The data is normalized to the average signal at centrosomes in control cells prior to cooling. Note how the centrosomal Jupiter-mCherry signal quickly drops on cooling and then immediately increases on warming in both control and *cnn,grip71,grip163* mutant cells, showing that centrosomes within both control and *cnn,grip71,grip163* mutant cells nucleate microtubules. **(G and H)** Graphs show the depolymerization (G) and nucleation/regrowth phases (H) phases from the graph in F. One-phase exponential decay models and "exponential plateau" models generated in GraphPad Prism using least squares fit are fitted to the depolymerization and nucleation/regrowth phases, respectively. The fits were compared using an extra sum-of-squares *F* test. Note how the centrosomal Jupiter-mCherry signal decreases faster upon cooling, but increases slower upon warming, in *cnn,grip71,grip163* mutant cells. **(I and J)** Fluorescent images documenting the behavior of the microtubule plus-end marker EB1-GFP within living *Drosophila* control (I) and *cnn,grip71,grip163* mutant (J) third instar larval brain cells as they were cooled to 5°C and then rapidly warmed to 20°C. Time in seconds relative to the initiation of warming (0 s) is indicated. Note how the EB1-GFP signal emanates from the centrosome and from the spindle/chromatin region during warming in the *cnn,grip71,grip163* mutant cell.

being nucleated elsewhere and then attaching to the centrosomes), we performed a cooling–warming microtubule repolymerization assay. We depolymerized microtubules by cooling larval brains on ice for ~40 min and then either chemically fixed samples on ice or allowed them to warm up for 30 s before rapid chemical fixation. ~40 min of cooling efficiently depolymerized microtubules at most centrosomes in wild-type and *cnn,grip71,grip163* mutant centrosomes (Fig. 3 B). After 30 s warming, all wild-type and *cnn,grip71,grip163* centrosomes had an associated α-tubulin signal, either as asters or as part of a re-formed mitotic spindle (Fig. 3 C), strongly suggesting that an accumulation of γ-tubulin at mitotic centrosomes is not necessary for these centrosomes to nucleate microtubules.

To better understand microtubule dynamics at wild-type and *cnn,grip71,grip163* centrosomes, we established a system to image cells live while cooling and warming the sample. We generated stocks containing fluorescent markers of microtubules (Jupiter-mCherry) and centrosomes (GFP-PACT) with and without the *cnn*, *grip71*, and *grip163* mutations and used a microscope-fitted heating–cooling device (CherryTemp) to modulate the temperature of larval brain samples during recording. We imaged samples for ~30 s before cooling them to 5°C for 3 min to depolymerize microtubules and then rapidly warming them to 20°C to observe microtubule regrowth. When cells were cooled to 5°C, the centrosomal Jupiter-mCherry signal decreased toward cytosolic background levels at both control and *cnn,grip71,grip163* centrosomes (Fig. 3, D–F; and Videos 1 and 2). In a subset of cells, this centrosomal signal reached cytosolic levels (i.e., disappeared) after 3 min of cooling (Fig. S2, A and B). On warming to 20°C, there was an immediate increase in the centrosomal Jupiter-mCherry signal at all control and *cnn,grip71,grip163* centrosomes (Fig. 3, D–F; and Fig. S2, A and B), confirming that microtubules can be nucleated at mitotic centrosomes that have not accumulated γ-tubulin. The dynamics of the microtubules differed, however (Fig. 3 F). On cooling to 5°C, microtubules depolymerized faster at control centrosomes (Fig. 3 F)—fitting "one-phase exponential decay" models to the depolymerization phases produced half-lives of 21.79 and 44.88 s and decay rate constants of 0.0361 and 0.0154 for control and *cnn,grip71,grip163* centrosomes, respectively. On warming to 20°C, microtubules also polymerized faster at control centrosomes (Fig. 3 F)—fitting "exponential plateau" models produced growth rate constants of 0.0759 and 0.0536, respectively,

which when normalized to the YM values (the maximum plateau values) showed an ~3.4-fold difference in growth rate (Fig. 3 H). Differences in microtubule dynamics were also apparent when imaging EB1-GFP comets, which mark growing microtubule plus ends. EB1-GFP comets emerging from control centrosomes disappeared immediately after cooling to 5°C and then reappeared immediately after warming to 20°C (Fig. 3 I and Video 3), but comets emerging from mutant centrosomes took longer to disappear, and fewer reappeared, during cooling and warming cycles (Fig. 3 J and Video 4). Moreover, it was easier to observe EB1-GFP comets emerging from chromatin regions in these *cnn,grip71,grip163* mutant cells (Fig. 3 J and Video 4), presumably because the centrosomes were no longer such dominant sites of microtubule nucleation. Thus, microtubules depolymerize faster and are then nucleated and/or polymerized faster at control centrosomes compared to at *cnn,grip71,grip163* centrosomes.

One caveat with the experiments above is that centrosome assembly is strongly perturbed in cells lacking the centrosome scaffold protein Cnn (Lucas and Raff, 2007; Conduit et al., 2014), potentially impacting the γ-TuRC-independent ability of centrosomes in *cnn,grip71;grip163* mutant cells to nucleate and organize microtubules. We, therefore, generated *cnn^{ΔCM1},grip71,grip163* mutants with or without GFP-PACT and Jupiter-mCherry, allowing us to examine microtubule dynamics at mitotic centrosomes that did not accumulate γ-tubulin (Fig. 4, A and B) but that still had Cnn to help assemble the PCM (although we note that PCM assembly appears perturbed to some degree in Cnn^{ΔCM1} mutant cells—see Fig. S1, E and F). Prior to cooling, centrosomes in *cnn^{ΔCM1},grip71,grip163* mutant cells had a Jupiter-mCherry signal that was on average slightly higher than in control cells, suggesting robust microtubule organization (Fig. 4, C–E). Similar to *cnn,grip71,grip163* mutants, microtubules depolymerized slower at *cnn^{ΔCM1},grip71,grip163* centrosomes compared with controls (Fig. 4, C–E). Fitting models to the data revealed half-lives of 16.7 and 37.7, and decay rate constants of 0.0416 and 0.0184 for control and *cnn^{ΔCM1},grip71,grip163* mutants, respectively (Fig. 4 G). However, the signal plateaued at a relatively high value despite cooling for 5 min as opposed to 3 (Fig. 4 E), suggesting that a larger proportion of microtubules are cold-stable at *cnn^{ΔCM1},grip71,grip163* mutant centrosomes compared with *cnn,grip71,grip163* mutant and control centrosomes. On warming to 20°C, microtubules polymerized at *cnn^{ΔCM1},grip71,grip163* centrosomes but again at a slower

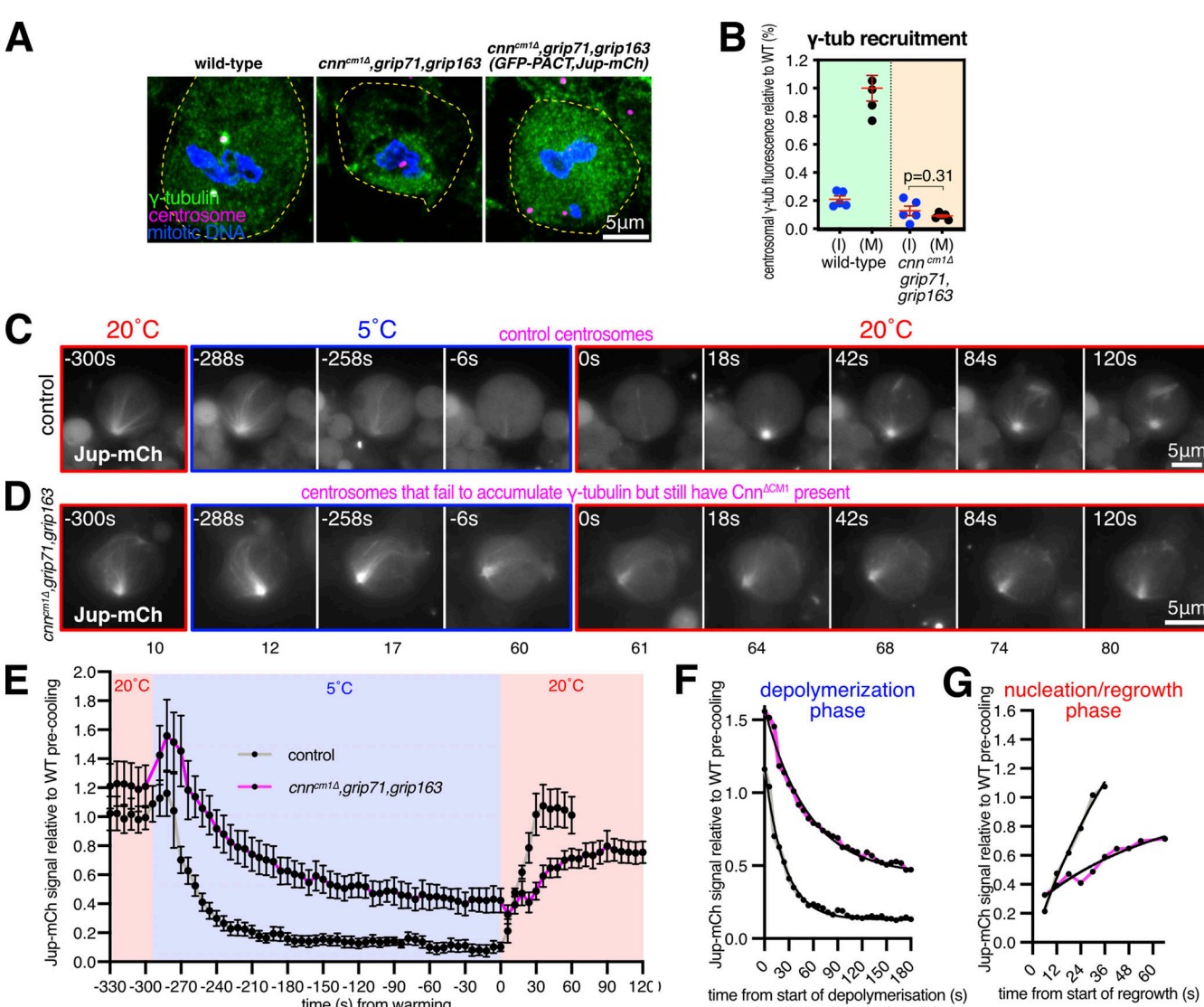

Figure 4. **Mitotic centrosomes that fail to accumulate γ-tubulin nucleate microtubules that are cold-resistant. (A–C)** Fluorescence images of mitotic *Drosophila* brain cells from either wild-type third instar larvae, $cnn^{\Delta CM1}$,*grip71*,*grip163* mutant third instar larvae, or $cnn^{\Delta CM1}$,*grip71*,*grip163* mutant third instar larvae also expressing GFP-PACT and Jupiter-mCherry, immunostained for γ-tubulin (green), mitotic DNA (blue), and Asl (centrioles, magenta). Note that GFP and mCherry fluorescence signals are destroyed during the fixation process due to the addition of acetic acid. Scale bar is 5 μm and applies to all images. **(B)** Graph showing average centrosomal fluorescence intensities of γ-tubulin (relative to wild-type) of interphase (blue dots) and mitotic (black dots) centrosomes from either wild-type or $cnn^{\Delta CM1}$,*grip71*,*grip163* mutants. Each datapoint represents the average centrosome value from one brain. N = 4 for WT and 5 for $cnn^{\Delta CM1}$,*grip71*,*grip163* for both interphase and mitosis. Mean and SEM are indicated. A two-sided paired *t* test was used to compare mean values of interphase and mitotic centrosomes, showing that there is no accumulation of γ-tubulin at mitotic centrosomes within the $cnn^{\Delta CM1}$,*grip71*,*grip163* mutant genotype. **(C–E)** Fluorescent images (C and D) and graph (E) documenting the behavior of Jupiter-mCherry within living *Drosophila* control (C) or $cnn^{\Delta CM1}$,*grip71*,*grip163* mutant (D) third instar larval brain cells as they were cooled to 5°C for 5 min and then rapidly warmed to 20°C. Time in seconds relative to the initiation of warming (0 s) is indicated. Note that the GFP-PACT signal used to locate centrosomes is not displayed. The graph in E plots the mean and SEM centrosomal signal (after subtraction of cytosolic signal) of 12 and 11 centrosomes from 8 and 9 control and $cnn^{\Delta CM1}$,*grip71*,*grip163* mutant cells, respectively. The data is normalized to the average signal at centrosomes in control cells prior to cooling. Note that a relatively large fraction of the centrosomal Jupiter-mCherry signal remains at centrosomes during cooling in $cnn^{\Delta CM1}$,*grip71*,*grip163* mutant cells, showing that the microtubule nucleated by these centrosomes are very cold-resistant. **(F and G)** Graphs show the depolymerization (F) and nucleation/regrowth phases (G) phases from the graph in E. One-phase exponential decay models and "exponential plateau" models generated in GraphPad Prism are fitted. The fits were compared using an extra sum-of-squares F test. Note how the centrosomal Jupiter-mCherry signal decreases faster upon cooling, but increases slower upon warming, in $cnn^{\Delta CM1}$,*grip71*,*grip163* mutant cells.

rate than at control centrosomes (Fig. 4, C–E): growth rate constants normalized to the YM values showed an ~3.4-fold difference in growth rate (Fig. 4 H), very similar to the growth rate for *cnn*,*grip71*,*grip163* mutant centrosomes. The absence of increased microtubule nucleation from $cnn^{\Delta CM1}$,*grip71*,*grip163* centrosomes

may in part reflect the fact that Cnn$^{\Delta CM1}$ does not appear to support PCM assembly as well as wild-type Cnn (Fig. S1 E). Moreover, we note that the absence of Grip71 may impact the ability of Augmin to amplify the microtubules being nucleated from centrosomes, thereby reducing nucleation efficiency compared with controls.

We conclude that centrosomes can nucleate microtubules independently of γ-TuRCs, but these microtubules are nucleated slower or grow slower and are more cold-stable than microtubules nucleated from wild-type centrosomes. This suggests that different modes of microtubule nucleation generate microtubules with different properties.

**The TOG domain protein Msps promotes microtubule nucleation from centrosomes lacking γ-tubulin complexes**

We next addressed which proteins promote γ-TuRC-independent microtubule nucleation at mitotic centrosomes. We did not observe any clear enrichment of α-tubulin at centrosomes after microtubule depolymerization (Fig. 3 B and Fig. S3), ruling out the possibility that a high local concentration of α/β-tubulin accounts for or contributes to γ-TuRC-independent microtubule nucleation. Proteins of the chTOG/XMAP215 and TPX2 protein families have been reported to promote γ-TuRC-independent microtubule nucleation. These proteins promote microtubule nucleation in a range of species both in vitro and in vivo, including in the absence of γ-TuRCs (see Discussion and references therein). The *Drosophila* homolog of chTOG is Minispindles (Msps), which binds microtubules, localizes to centrosomes and spindle microtubules, and is required for proper spindle formation, mitotic progression, and chromosome segregation (Cullen et al., 1999). Msps has also been reported to stabilize the minus ends of microtubules when bound and recruited to centrosomes by TACC (Barros et al., 2005; Lee et al., 2001). Msps is also part of a group of proteins that organize microtubules independently of γ-tubulin at the nuclear envelope of fat body cells (Zheng et al., 2020). Moreover, the TOG1 and TOG2 domains of Msps promote microtubule nucleation in vitro (Slep and Vale, 2007). The putative *Drosophila* TPX2 homolog is Mei-38 and, while its depletion results in only mild spindle defects, Mei-38 binds microtubules, localizes to centrosomes and spindle microtubules, and is required for microtubule regrowth from kinetochores (Popova et al., 2022; Goshima, 2011). CAMSAP/Patronin/Nezha protein family members have also been implicated in γ-TuRC-independent microtubule nucleation and organization at non-centrosomal sites (Akhmanova and Kapitein, 2022), and CAMSAP2 condensates can stimulate microtubule nucleation in vitro (Imasaki et al., 2022).

To test the role of these proteins in γ-TuRC-independent nucleation from centrosomes, we combined mutant or RNAi alleles with the *cnn*, *grip71*, and *grip163* mutant alleles and analyzed microtubule organization at centrosomes during prophase, when microtubule asters are most robust (Conduit et al., 2014). We were unable to obtain third instar larvae when combining the *cnn*, *grip71*, and *grip163* mutant alleles with *patronin* mutant or RNAi alleles, presumably due to severe microtubule defects that prevented development, and thus could not test the role of Patronin. We could, however, obtain larvae when combining the *cnn*, *grip71*, and *grip163* mutant alleles with mutant alleles for *msps* or *tacc* or an RNAi allele for Mei38. A clear association of microtubules with centrosomes was observed in 100% of wild-type prophase cells and in 96.8% of *cnn,grip71,grip163* mutant prophase cells (Fig. 5, A, B, and F), consistent with our observations above that *cnn,grip71,grip163* centrosomes can nucleate

and organize microtubules. In contrast, a clear association of microtubules with centrosomes was observed in only 55.3% of *cnn,grip71,grip163,msps* mutant prophase cells, and in 70.7% and 81.4% of *cnn,grip71,grip163,tacc* and *cnn,grip71,grip163,mei-38-RNAi* mutant cells, respectively (Fig. 5, C–F). Moreover, the *cnn,grip71,grip163,msps* centrosomes tended to be positioned further from the spindle poles than the *cnn,grip71,grip163* centrosomes (Fig. 5, G and H), which is indicative of a reduced capacity to organize microtubules. Positioning of centrosomes in *cnn,grip71,grip163,tacc* and *cnn,grip71,grip163,mei-38-RNAi* mutant cells was less affected, presumably due to the less severe defects in microtubule organization at *cnn,grip71,grip163,tacc* and *cnn,grip71,grip163,Mei38-RNAi* centrosomes.

Given that Msps appeared to be most important for γ-TuRC-independent nucleation of microtubules from centrosomes, we tested its role directly by performing a cooling/warming microtubule nucleation assay (similar to the fixed cell assay performed in Fig. 3, B and C) and compared the recovery of microtubules at *cnn,grip71,grip163* and at *cnn,grip71,grip163,msps* centrosomes 30 s after warming. We categorized cells as those with or without centrosomes (some cells lack centrosomes due to mis-segregation during mitosis) and those that had or had not yet formed spindles; the proportion of these categories was similar in both mutant types (Fig. S4). There were, however, differences between the mutant types within each category. Of the cells that contained centrosomes but had not yet formed a spindle, centrosomes organized microtubules in ~94.3% of *cnn,grip71,grip163* mutant cells, the majority of which were scored as having strong or medium asters, but centrosomes organized microtubules in only ~37.7% of *cnn,grip71,grip163,msps* mutant cells, the majority of which were scored as having weak asters (Fig. 5, I and J). This difference appeared to affect spindle formation because, of the cells that had centrosomes and that had formed a spindle structure, spindles were scored as being of "high" or "medium" quality (based on their morphology and density) in ~67.1% of *cnn,grip71,grip163* mutant cells but in only ~28% of *cnn,grip71,grip163,msps* mutant cells (Fig. S5, A and B). This was specific to centrosomes because there was a similarly high proportion of cells containing low-quality spindles in both mutant types when cells lacked centrosomes (Fig. S5 C). For comparison, spindles were scored as being of "high" or "medium" quality in ~95.3% of wild-type cells (Fig. S5 A). Note also that the absence of Grip71 abolishes the Augmin-mediated nucleation pathway necessary for efficient spindle assembly (Reschen et al., 2012; Chen et al., 2017b; Dobbelaere et al., 2008; Vérollet et al., 2006), but as both mutant types lacked Grip71 this cannot explain the differences observed between the mutants.

In summary, our data show that centrosomes lacking γ-tubulin complexes can still nucleate microtubules, despite having reduced PCM, and that the TOG domain protein Msps plays an important role in this γ-TuRC-independent microtubule nucleation pathway.

## Discussion

How centrosomes nucleate and organize microtubules is a longstanding question. Centrosomes contain hundreds of proteins,

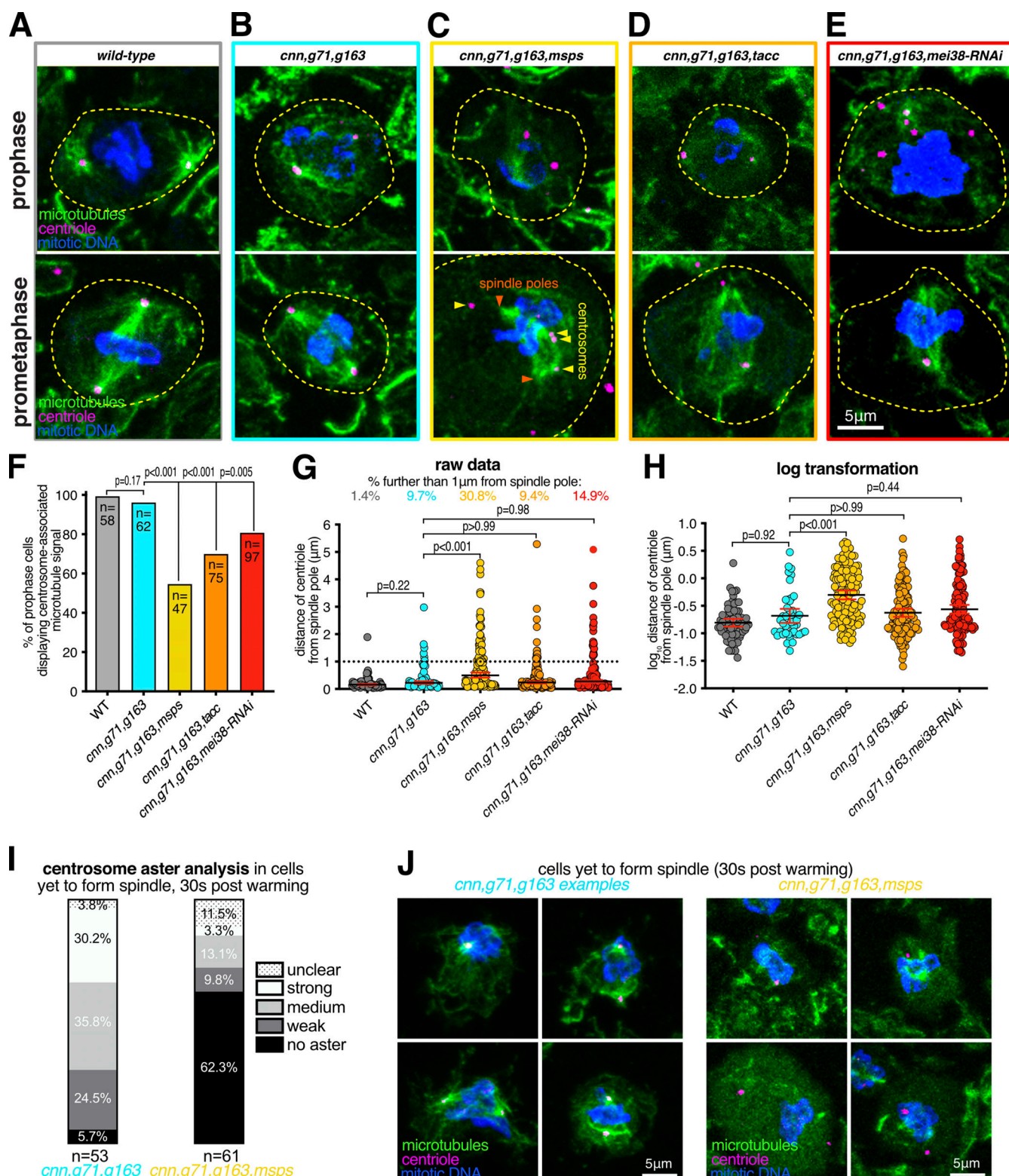

Figure 5. **Depletion of Msps strongly perturbs the ability of centrosomes to organize and nucleate microtubules in the absence of γ-tubulin complexes. (A–E)** Fluorescence images of *Drosophila* brain cells in either prophase or prometaphase from either wild-type (A), *cnn,grip71,grip163* (B), *cnn,grip71,grip163,msps* (C), *cnn,grip71,grip163,tacc* (D), or *cnn,grip71,grip163,mei38-RNAi* cell (E) immunostained for alpha-tubulin (microtubules, green), mitotic DNA (blue), and Asl (centrioles, magenta). Note that some cells lacking Cnn have abnormal numbers of centrosomes due to centrosome segregation problems during cell division (Conduit et al., 2010). **(F)** Graph showing the percentage of prophase cells in which microtubules are associated with at least one centrosome within the various genotypes, as indicated. The number of cells analyzed (*n*) is indicated. Datasets were compared to the *cnn,grip71,grip163* dataset using one-way Chi-squared tests. Note there is a no significant reduction between wild-type and *cnn,grip71,grip163* mutant cells in the proportion of cells displaying centrosome-associated microtubules, but there are significant reductions between *cnn,grip71,grip163* mutant cells and cells that are also depleted for

either Msps, TACC, or *mei-38*, indicating that Msps, TACC, and *mei-38* have a role in γ-TuRC-independent microtubule nucleation. **(G and H)** Graphs of raw data (G) and log transformed data (H) showing the distance of centrosomes from spindle poles during prometaphase in the different genotypes, as indicated. The percentage of centrosomes that were farther than 1 μm from the spindle poles is indicated above each dataset in the graph in G. Increased distance from the spindle pole is indicative of a failure to properly organize microtubules. Kruskal-Wallis tests were used to compare the distribution of the *cnn,grip71,grip163* dataset with those of the other genotypes. Each datapoint represents an individual centrosome. N numbers are as: WT 62, *cnn,grip71,grip163* 72, *cnn,grip71,-grip163,msps* 133, *cnn,grip71,grip163,tacc* 139, and *cnn,grip71,grip163,mei38-RNAi* 134. Note that there is a significant difference only between *cnn,grip71,grip163* and *cnn,grip71,grip163,msps* mutant cells, indicating that Msps is particularly important for microtubule organization at centrosomes in the absence of γ-TuRCs. **(I and J)** Parts of a whole graph (I) and images (J) represent analyses of centrosomal aster types in cells fixed and immunostained for alpha-tubulin (microtubules, green), mitotic DNA (blue), and Asl (centrioles, magenta) after 30 s of warming post cooling from either *cnn,grip71,grip163* or *cnn,grip71,grip163,msps* mutants, as indicated. *N* numbers are indicated, and each N represents a single cell. Only cells that had centrosomes but that had not yet formed a spindle were analyzed. Note how centrosome asters are frequently absent in *cnn,grip71,grip163,msps*.

---

many of which associate with microtubules, meaning that understanding how centrosomes nucleate and organize microtubules is not trivial. Prior to our current work, we had identified Cnn and Spd-2 as the two key PCM components in flies—remove one and the other could support partial PCM assembly and microtubule organization; remove both and PCM assembly and microtubule organization fail (Conduit et al., 2014). We found that γ-tubulin could still accumulate at mitotic centrosomes after the removal of either Cnn or Spd-2, showing that both proteins could mediate the recruitment of γ-tubulin complexes, but it remained unclear how. The work presented here shows that Cnn and Spd-2 recruit different types of γ-tubulin complex, with Cnn able to recruit γ-TuSCs and Spd-2 recruiting predominantly preformed γ-TuRCs. Moreover, by preventing γ-tubulin recruitment but not PCM assembly, we have shown that centrosomes still nucleate microtubules in the absence of γ-TuRCs and that this γ-TuRC-independent mode of microtubule nucleation is stimulated by the TOG domain protein Msps (Fig. 6).

By using classical genetics, we have found that Cnn can recruit γ-tubulin complexes independently of Grip71 and the GCP4/5/4/6 core, meaning that it must be able to recruit γ-TuSCs. This is consistent with previous observations showing that γ-TuSCs could still be recruited to mitotic *Drosophila* centrosomes in S2 cells lacking the GCP4/5/4/6 core components (Vogt et al., 2006), although it was unknown at that time that this recruitment was dependent on Cnn. Our data here also show that this occurs in vivo. The ability of Cnn to recruit γ-TuSCs is similar to its budding yeast homologs' ability, where Grip71 and the GCP4/5/4/6 core are naturally absent. Consistent with this, we show that Cnn's binding and recruitment of γ-tubulin complexes relies entirely on its highly conserved CM1 domain, which binds across the inter γ-TuSC interface in budding yeast complexes (Brilot et al., 2021). The binding of the CM1 domain in budding yeast stimulates the oligomerization of γ-TuSCs into γ-TuRCs (Kollman et al., 2010; Lyon et al., 2016; Brilot et al., 2021; Lin et al., 2014; Gunzelmann et al., 2018), but whether this is true of Cnn's CM1 domain, or CM1 domains in other eukaryotes, remains to be determined. Consistent with this possibility, however, Cnn's CM1 domain is more similar to Spc110's rather than Spc72's CM1 domain, which unlike Spc72 does not require the TOG domain protein Stu2 for efficient oligomerization of γ-TuSCs. Moreover, Grip71 and the GCP4/5/4/6 core components are neither essential in flies (Reschen et al., 2012; Vogt et al., 2006) nor in *Aspergillus* or *S. pombe* (Xiong and

Oakley, 2009; Anders et al., 2006), suggesting that there must be ways to assemble ring-like templates in these organisms in the absence of the GCP4/5/4/6 core. We speculate that this "other way" is via CM1-mediated oligomerization of γ-TuSCs (Fig. 6). Nevertheless, Cnn and other CM1 domain proteins can also bind γ-TuRCs formed via the GCP4/5/4/6 core (Muroyama et al., 2016; Choi et al., 2010; Tovey et al., 2021; Wieczorek et al., 2019, 2020) and so it remains unclear whether Cnn recruits γ-TuSCs only in the absence of preformed γ-TuRCs.

In contrast to Cnn, Spd-2 (which does not contain a CM1 domain) requires Grip71 and the GCP4/5/4/6 core to recruit γ-tubulin complexes to mitotic centrosomes i.e., it is unable to bind and recruit γ-TuSCs directly or mediate their recruitment by another protein. Whether Spd-2 binds directly to preformed γ-TuRCs remains unclear. Grip71 associates with preformed γ-TuRCs in the cytosol and the human homolog of Grip71, NEDD1, has been reported to interact with the human homolog of Spd-2, CEP192 (Gomez-Ferreria et al., 2012a). Thus, Spd-2 might recruit γ-TuRCs via binding to Grip71, but since we show that Spd-2 can recruit γ-TuRCs in the absence of Grip71, it must also be able to recruit γ-TuRCs in a different way. Our data show that Grip163[GCP6] is more important than Grip75[GCP4] in this respect because removing Grip71 and Grip75[GCP4] does not completely abolish γ-tubulin accumulation. Perhaps, therefore, Spd-2 recruits γ-TuRCs via an interaction with Grip163[GCP6]: or alternatively Grip163[GCP6], but not Grip75[GCP4], is essential for the assembly of preformed γ-TuRCs that are in turn necessary for Spd-2-mediated recruitment. This latter possibility would be consistent with findings in human cells, where depletion of GCP6 is more disruptive to γ-TuRC assembly (Cota et al., 2017). So far, our attempts to identify direct interactions between Spd-2 and γ-TuRC components have failed, so it is possible that an intermediary protein links Spd-2 to γ-TuRCs.

The finding that Cnn and Spd-2 recruit different types of γ-tubulin complexes to centrosomes fits well with recent observations that not all γ-TuRCs within a given species or cell type have the same protein composition. This was shown by analyzing the γ-TuRC protein Mzt1 in *Drosophila, Caenorhabditis elegans*, fission yeast, and *Aspergillus*, where Mzt1 is either not present or not necessary at certain MTOCs (Tovey et al., 2018; Gao et al., 2019; Huang et al., 2020; Sallee et al., 2018). For example, we have shown that *Drosophila* Mzt1 is expressed only in developing sperm cells and is required for γ-TuRC recruitment to basal bodies but not mitochondria (Tovey et al., 2018). Moreover, in mouse epithelial cells, γ-TuRCs are bound and

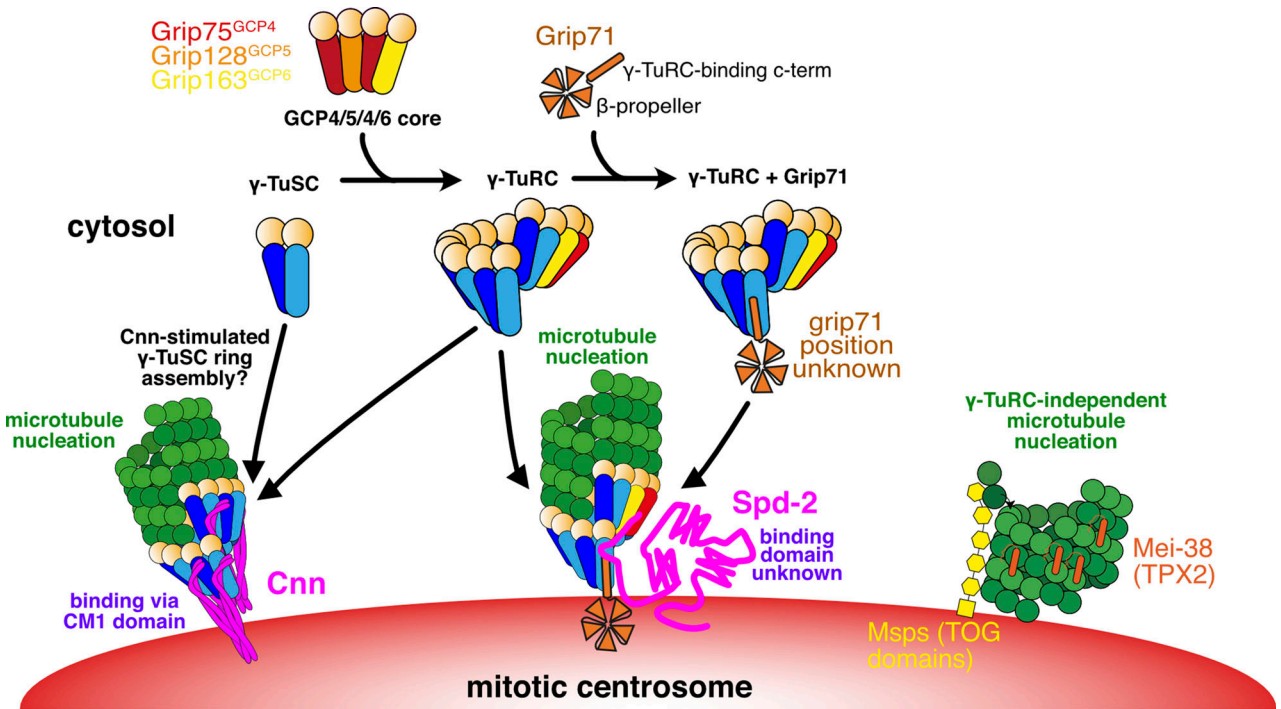

Figure 6.   **Model for the different pathways of γ-tubulin complex recruitment and microtubule nucleation at mitotic centrosomes in *Drosophila*.** This model is based on both previous data from the literature and our current findings. A mixture of γ-TuSCs and γ-TuRCs exists in the cytosol. The GCP4/5/4/6 core, predicted to comprise Grip75GCP4, Grip128GCP5, and Grip163GCP6 in *Drosophila*, is necessary for γ-TuSCs to assemble into γ-TuRCs within the cytosol. Grip71 is a peripheral γ-TuRC protein that can associate with cytosolic γ-TuRCs but is not necessary for their assembly and so cytosolic γ-TuRCs with and without Grip71 may exist. Cnn is able to recruit γ-tubulin in the absence, or at least near absence, of the GCP4/5/4/6 core and Grip71, suggesting it can recruit γ-TuSCs directly from the cytosol. It likely also recruits pre-formed γ-TuRCs under normal conditions because artificial Cnn scaffolds recruit Grip75GCP4-GFP (Tovey et al., 2021). Cnn's ability to recruit γ-tubulin complexes relies on its highly conserved N-terminal CM1 domain. We speculate that CM1 domain binding may stimulate γ-TuSC oligomerization into γ-TuSC-only γ-TuRCs that could then nucleate microtubules, as is true of CM1 domain proteins in yeast. In contrast to Cnn, Spd-2 recruitment relies largely on the GCP4/5/4/6 core and so Spd-2 must predominantly recruit pre-formed γ-TuRCs. Spd-2 may be able to recruit very low levels of γ-TuSCs via Grip71 (not depicted). How Spd-2 binds to γ-tubulin complexes remains unknown. When the recruitment of γ-tubulin complexes by both Cnn and Spd-2 is inhibited, centrosomes are still able to nucleate microtubules and this γ-tubulin-independent microtubule nucleation pathway depends on Msps (the fly TOG domain protein), and possibly Mei-38 (the putative homolog of TPX2). Note that these proteins also likely facilitate γ-TuRC-dependent nucleation (not depicted).

recruited either by CDK5RAP2 (Cnn homolog) or by NEDD1 (Grip71 homolog), and this influences the nucleation and anchoring ability of the γ-TuRCs (Muroyama et al., 2016). Whether other forms of γ-TuRCs also exist and how this affects their function remains to be explored.

In addition to revealing details of centrosomal recruitment of γ-tubulin complexes, we have also shown that microtubules can be nucleated in their apparent absence. We've known for some time that microtubules are present within cells after depletion of γ-tubulin or other key γ-TuRC proteins (Hannak et al., 2002; Strome et al., 2001; Sampaio et al., 2001; Sunkel et al., 1995; Tsuchiya and Goshima, 2021; Sallee et al., 2018; Rogers et al., 2008; Nakaoka et al., 2015; Wang et al., 2015; Gunzelmann et al., 2018) and that certain non-centrosomal MTOCs naturally lack γ-tubulin (Nashchekin et al., 2016; Yang and Wildonger, 2020; Zheng et al., 2020; Mukherjee et al., 2020; Kitamura et al., 2010). Mounting evidence, including our work here, suggests that the ch-Tog/XMAP215/Msps/Alp14/Stu2 TOG domain family of proteins (which have microtubule polymerase activity) and the TPX2 family of proteins (which have microtubule stabilization activity) are important for microtubule nucleation.

Depletion of TOG domain proteins from *Xenopus* egg extracts, *Drosophila* S2, fat body cells, fission yeast cells, and budding yeast cells, and depletion of TPX2 from *Xenopus* egg extracts severely impairs microtubule nucleation or organization (Popov et al., 2002; Thawani et al., 2018; Zheng et al., 2020; Groen et al., 2009; Flor-Parra et al., 2018; Gunzelmann et al., 2018; Rogers et al., 2008). TOG domain and TPX2 proteins have been shown to work together with γ-TuRCs (or microtubule seed templates) to promote microtubule nucleation (Thawani et al., 2018; Flor-Parra et al., 2018; Gunzelmann et al., 2018; Consolati et al., 2020; King et al., 2020; Wieczorek et al., 2015). Consistent with this, codepletion of γ-tubulin and the *Drosophila* TOG domain protein Msps did not delay non-centrosomal microtubule regrowth after cooling compared with single depletions in interphase S2 cells (Rogers et al., 2008). Nevertheless, several studies, mainly in vitro, have shown that TOG and TPX2 proteins can also function independently of γ-TuRCs to promote microtubule nucleation (Roostalu et al., 2015; Woodruff et al., 2017; Schatz et al., 2003; Slep and Vale, 2007; Ghosh et al., 2013; Thawani et al., 2018; King et al., 2020; Zheng et al., 2020; Tsuchiya and Goshima, 2021). Our data suggest that, unlike from non-

centrosomal sites in interphase S2 cells, Msps can promote γ-TuRC-independent microtubule nucleation from centrosomes in mitotic larval brain cells. This difference may reflect Msps having a high local concentration at centrosomes. This finding is similar to that of a recent study in human colon cancer cells showing that γ-tubulin depletion did not prevent microtubule nucleation from centrosomes and that this γ-TuRC-independent microtubule nucleation pathway depended on the Msps homolog ch-TOG (Tsuchiya and Goshima, 2021). It also supports the observation that *C. elegans* centrosome-like condensates nucleate microtubules with help from the TOG domain protein Zyg-9 (Woodruff et al., 2017). It is possible that Patronin is also involved in γ-TuRC-independent microtubule nucleation from centrosomes, but we were unable to test this. We note, however, that endogenously tagged Patronin-GFP is not readily detectable at mitotic centrosomes in larval brain cells (unpublished observations). Interestingly, α/β-tubulin does not concentrate at mitotic centrosomes in flies (Fig. S3), unlike in *C. elegans,* where this can promote microtubule nucleation (Woodruff et al., 2017; Baumgart et al., 2019).

So why are γ-TuRCs required at all? While microtubules can be nucleated independently of γ-TuRCs, nucleation or microtubule growth appears to be more efficient when γ-TuRCs are present (Tsuchiya and Goshima, 2021; Hannak et al., 2002; this study). Naturally occurring γ-TuRC-independent microtubule nucleation at specialized MTOCs, such as the nuclear envelope of *Drosophila* fat body cells (Zheng et al., 2020), may not require a high frequency of microtubule nucleation events, perhaps because they build their microtubule arrays over a relatively long period of time. During cell division, however, many microtubules must be generated rapidly, possibly creating a requirement for γ-TuRCs to provide efficient microtubule nucleation. Indeed, depleting γ-TuRCs delays spindle assembly and results in spindle and chromosome defects (Sunkel et al., 1995; Sampaio et al., 2001; Colombié et al., 2006; Vérollet et al., 2006). Nevertheless, centrosomes lacking γ-TuRCs can organize similar, if not higher, numbers of microtubules as in controls (as observed in *cnn^cm1,grip71,grip163* mutant cells). These microtubules are, however, less dynamic, being more cold-resistant, and so perhaps going through rounds of depolymerization/polymerization less frequently. This reduced dynamicity may impact spindle assembly. In addition, γ-TuRCs may also be important to set microtubule protofilament number and define microtubule polarity, and studies have implicated γ-tubulin or γ-TuRC proteins in the control of microtubule dynamics and of cell cycle progression, independent of their microtubule nucleation roles (Oakley et al., 2015; Bouissou et al., 2009).

In summary, our data highlight the robustness of centrosomes to nucleate microtubules. We have shown that centrosomes can recruit different forms of γ-tubulin complexes (γ-TuSCs and γ-TuRCs) via multiple pathways and that they can nucleate and organize microtubules in the absence of γ-tubulin complexes. This γ-TuRC-independent mode of microtubule nucleation relies on the TOG domain protein Msps. This multipathway redundancy helps explain why centrosomes are such dominant MTOCs during mitosis. A seemingly important finding is that microtubules nucleated by different mechanisms have

different properties. This concept is similar to how the plus-end dynamics of yeast microtubules are a function of where the microtubules were nucleated (Chen et al., 2019). These unexpected observations deserve further investigation.

## Materials and methods

### Transgenic and endogenously modified *Drosophila* lines
The Jupiter-mCherry line was generated previously (Lu et al., 2013), but the details of its generation were not described. Nevertheless, it is a widely used fly line reported in FlyBase and has been used previously by us (Conduit et al., 2014). GFP-PACT (Martinez-Campos et al., 2004) and RFP-PACT (Conduit et al., 2010) lines were generated previously by using Gateway cloning (Thermo Fisher Scientific) to insert the sequence encoding the PACT domain of *Drosophila* Pericentrin-Like Protein (D-PLP) into a pUbq-GFP or p-Ubq-RFP vector containing a p-element and a the mini-white gene to allow random insertion into the *Drosophila* genome. To delete the CM1 domain from Cnn, we first designed a pCFD4 vector (Port et al., 2014) containing two guide RNAs with the following target sequences: 5′-AACTCGCCCTTG CCGTCACA-3′ and 5′-GTGATGAGAAATGGCTCGAG-3′. This vector was injected into flies containing the attP2 landing site by Rainbow Transgenic Flies, Inc. Male flies were then crossed to females expressing nos-cas9 (BL54591) and the resultant embryos were injected by the Department of Genetics Fly facility, University of Cambridge, UK, with a homology vector encoding 1 kb on either side of the deletion region (R98 to D167, inclusive) and including silent mutations to disrupt the guide RNAs. The resulting F0 flies were crossed to balancer lines and their progeny were screened by PCR for the deletion using "amplification primer 1": 5′-ATTGGATGTTGTGCTGCGAGG-3′ and "amplification primer 2": 5′-TTCAGATAAGTGTCGTGCTCG-3′. Sequencing of the PCR product was performed by Eurofins using "amplification primer 2" (Table 1).

The endogenously-tagged EB1-GFP line was made using CRISPR-based genome editing by inDroso, France. An SSSS-eGFP-3′UTR-LoxP-3xP3-dsRED-LoxP cassette was inserted and then the selection markers were excised. The guide RNA sequences were not communicated, and the company has now closed.

### Mutant alleles, RNAi lines, and fly stocks
Wild-type flies used in the study were $w^{1118}$. For *spd-2* mutants, we used the *dspd-2^{Z3571l}* mutant allele, which carries an early stop codon resulting in a predicted 56aa protein. Homozygous *dspd-2^{Z3571l}* mutant flies lack detectable Spd-2 protein on Western blots and so the allele is considered a null mutant (Giansanti et al., 2008). In our stock collection, this allele no longer produces homozygous flies (which is common for mutant alleles kept as balanced stocks for many years), so we combined *dspd-2^{Z3571l}* with a deficiency that deletes the entire *spd-2* gene (*dspd-2^{Df(3L)st-j7}*). On Western blots, there was no detectable Spd-2 protein in brain extracts from flies carrying the *dspd-2^{Z3571l}/dspd-2^{Df(3L)st-j7}* hemizygous mutations (Fig. S6 B). For *cnn* mutants, we combined the *cnn^{f04547}* and *cnn^{HK21}* mutant alleles. The *cnn^{f04547}* allele carries a piggyBac insertion in the middle of the cnn gene and is

Table 1. **Primers**

| Experiment type | Vector or construct | Forward primer | Reverse primer |
|---|---|---|---|
| CM1 deletion | pCFD4 guide RNA vector | CFD4_CM1_1f 5′-TATATAGGAAAGATATCCGGGTGAACTTCGAACTCGCCCTTGCCGTCACAGTTTTAGAGCTAGAAATAGCAAG-3′ | CFD4_CM1_3b 5′-ATTTTAACTTGCTATTTCTAGCTCTAAAACCTCGAGCCATTTCTCATCACGACGTTAAATTGAAAATAGGTC-3′ |
| | Homology vector | CM1Del_BS_b1 5′-TAGCGATCGCTGATTTGGAACAGTCCGTAATCCCGGGATCCAATTCGCCCTATAGTG-3′ | CM1Del_BS_f1 5′-CATGACGGCGGATGCCGGGGTTGGTATCACATCTTCTCTTCAGCTTTTGTTCCCTTTAGTGAGGG-3′ |
| | | CM1Del_UHA_f1 5′-GATCCCGGGATTACGGACTGTTCCAAATCAG-3′ | CM1Del_UHA_b1 5′-CGTGATGAGAAATGGCTCGAGCTGCGCCTTGCGAGGGCAAGGGCGAGTTGCCGCCG-3′ |
| | | CM1Del_DHA_f1 5′-GCAGCTCGAGCCATTTCTCATCACG-3′ | CM1Del_DHA_b1 5′-GCTGAAGAGAAGATGTGATACCAACCCCGG-3′ |
| | PCR screening | Amplification primer 1 5′-ATTGGATGTTGTGCTGCGAGG-3′ | Amplification primer 2 5′-TTCAGATAAGTGTCGTGCTCG-3′ |
| Recombinant proteins | pDEST-MBP-PreSci-his-Strep | Empty pDEST_f 5′-AATTCGATCACAAGTTTGTACAAAAAAGC-3′ | Empty pDEST_mid_b 5′-TCTAAAGTATATATGAGTAAACTTGGTCTGACAGTTACCAATGCTTAATC-3′ |
| | | Empty pDEST_mid_f 5′-CAAGTTTACTCATATATACTTTAGATTGATTTACCCCGGTTGATAATC-3′ | His remove_b 5′-GATTTTCATAATCTATGGTCCTTGTTGGTGAAGTGCTCGTGAAAACACCTAAACGG-3′ |
| | | His remove_f 5′-CCAACAAGGACCATAGATTATGAAAATCGAAGAAGGTAAACTGGTAATCTGGATTAACGG-3′ | Empty pDEST_MBP_b 5′-AGTCTGCGCGTCTTTCAGGGCTTCATC-3′ |
| | | PreSci-His_Insert_f 5′-CGATGAAGCCCTGAAAGACGCGCAGACTAATTCGAGCCTGGAAGTTCTGTTCCAGGGGCCCAGTGGACATCACCATCACC-3′ | PreSci_His_Strep_Insert_b 5′-GCTTTTTTGTACAAACTTGTGATCGAATTACCTCCACTTTTCTCGAACTGCGGGTGGCTCCAAGTGTGATGGTGATGGTGATGTCCACTGGGCCCCTGG-3′ |
| | Cnn-C-N[T27E,F115A,S186D] | Cnn_F115A_QC_f 5′-CCGCGCTGCGCAAGGAGAACGCCAATCTAAAGCTGCGC-3′ | Cnn_F115A_QC_b 5′-GCGCAGCTTTAGATTGGCGTTCTCCTTGCGCAGCGCGG-3′ |
| | Cnn-C-N[T27E,R101Q,E102A,S186D] | CnnCM1_mut_QC_f1 5′-CCGTCACAGGGTCGCTCTGTACAGGCCTTGGAGGAGCAGATGTCC-3′ | CnnCM1_mut_QC_b1 5′-GGACATCTGCTCCTCCAAGGCCTGTACAGAGCGACCCTGTGACGG-3′ |

predicted to disrupt long Cnn isoforms, including the centrosomal isoform (Cnn-C or Cnn-PA; Lucas and Raff, 2007). This mutation is considered to be a null mutant for the long Cnn isoforms (Lucas and Raff, 2007; Conduit et al., 2014). The cnn[HK21] allele carries an early stop codon after Cnn-C's Q78 (Vaizel-Ohayon and Schejter, 1999) and affects both long and short Cnn isoforms—it is considered to be a null mutant (Eisman et al., 2009; Chen et al., 2017a). On Western blots, there was no detectable Cnn-C protein in brain extracts from flies carrying the cnn[f04547]/cnn[HK21] hemizygous mutations (Fig. S6 A). For Grip71, we used the grip71[120] mutant allele, which is a result of an imprecise p-element excision event that led to the removal of the entire grip71 coding sequence except for the last 12 bp; it is considered to be a null mutant (Reschen et al., 2012). We combined this with an allele carrying a deficiency that includes the entire grip71 gene (grip71[Df(2L)Exel6041]). On Western blots, there is no detectable Grip71 protein in grip71[120]/grip71[df6041] hemizygous mutant brains (see blots on the CRB website, which were performed by us). For Grip75[GCP4], we used the grip75[175] mutant allele, which carries an early stop codon after Q291. Homozygous grip75[175] mutant flies lack detectable Grip75[GCP4] protein on Western blots and so the allele is therefore considered to be a null mutant (Schnorrer et al., 2002). We combined this with an allele carrying a deficiency that deletes the entire grip75[GCP4] gene

(grip75[Df(2L)Exel7048]). In the absence of a working antibody, we have not confirmed the expected absence of Grip75[GCP4] protein in grip75[175]/grip75[Df(2L)Exel7048] hemizygous mutant flies on Western blots. For Grip128[GCP5], we used the UAS-controlled grip128-RNAi[V29074] RNAi line, which is part of the VDRC's GD collection—sequence: 5′-GCGCAAACGAAATATGGGAATGGAGGATGATTTGCTACTCGTGGAGATCTTCAACAAGCTGCAATCCTGCCCACTCTACCAGCTACTGCTGGAGCATGCCTTGGAGTCTGGCGAAACGCAAGATTTGCTATGTAGTGTAAATACGCTGAGCGAAATGCTGACCAGCAACAATGAAATCCAACTGCCGTCGCTGCACGATGAGCTGTTCACGCAGTTCTTTGCGCAGCTAAAGGTTTACTGTGGTGCGGACAACACGGATTACGAGGATGAGCCGGAGCCGGACAAAGACTACGAAGATCTGACTGTGTGCAATAGGCAGGGCATTAGGAACCATGAACTTTTCGCCATATTTACCCAGCCG-3′, and drove its expression using the Insc-Gal4 driver (BL8751), which is expressed in larval neuroblasts and their progeny. In the absence of a working antibody, we have not confirmed the absence or reduction of Grip128[GCP5] protein on Western blots. RNAi was used for grip128[GCP5] as its position on the X chromosome made generating stocks with multiple alleles technically challenging. For Grip163[GCP6], we used the grip163[GE2708] mutant allele, which carries a p-element insertion between amino acids 822 and 823 (total protein length is 1351aa) and behaves as a null or strong hypomorph mutant (Vérollet et al., 2006). We

combined this with an allele carrying a deficiency that deletes the entire *grip163*$^{GCP6}$ gene (*grip163*$^{Df(3L)Exel6115}$). In the absence of a working antibody, we have not confirmed the absence or reduction of Grip163$^{GCP6}$ protein in *grip163*$^{GE2708}$/*grip163*$^{Df(3L)Exel6115}$ hemizygous mutant flies on Western blots. For Msps, we used the *msps*$^{p}$ and *msps*$^{MJ15}$ mutant alleles. The *msps*$^{p}$ allele carries a p-element insertion within, or close to, the 5′ UTR of the *msps* gene and results in a strong reduction, but not elimination, of Msps protein on Western blots (Cullen et al., 1999). The *msps*$^{MJ15}$ allele was generated by remobilizing the p-element (the genetic consequence of which has not been defined) and also results in a strong reduction, but not elimination, of Msps protein on Western blots (Cullen et al., 1999; Lee et al., 2001). For TACC, we used the *tacc*$^{stella}$ allele, which contains a p-element insertion of unknown localization but that results in no detectable TACC protein on Western blots (Barros et al., 2005). For Mei-38, we used the UAS-controlled *mei-38*-RNAi$^{HMJ23752}$ RNAi line, which is part of the NIG's TRiP Valium 20 collection—sequence: CAGCCT GGAGCAGAAGAAGAA, and drove its expression using the Insc-Gal4 driver (BL8751). In the absence of a working antibody, we have not confirmed the absence or reduction of Mei-38 protein on Western blots. RNAi was used for *mei-38* as its position on the X chromosome made generating stocks with multiple alleles technically challenging. Moreover, the only available mutant of *mei-38* affects a neighboring gene.

For examining the behavior of MTs in living larval brain cells, we analyzed brains expressing two copies of Ubq-GFP-PACT and two copies of Ubq-Jupiter-mCherry in either a WT, a *cnn,grip71,grip163*$^{GCP6}$ mutant, or a *cnn*$^{ΔCM1}$,*grip71,grip163*$^{GCP6}$ mutant background. For examining the behavior of EB1-GFP in living larval brain cells, we analyzed brains expressing two copies of EB1-GFP in either a WT or a *cnn,grip71,grip163*$^{GCP6}$ mutant background.

### Antibodies
For immunofluorescence analysis, we used the following antibodies: mouse anti-γ-tubulin monoclonal (1:500; GTU88; Sigma-Aldrich), mouse anti-α-tubulin monoclonal (1:1,000; DM1α; Sigma-Aldrich), rabbit anti-α-tubulin monoclonal (1:500; AB52866; Abcam), anti-HistoneH3 (phospho-S10) mouse monoclonal (1:2,000, AB14955; Abcam), anti-HistoneH3 (phospho-S10) rabbit polyclonal (1:500, AB5176; Abcam), Guinea pig anti-Asl polyclonal (1:500; gift from Jordan Raff, Sir William Dunn School of Pathology, Oxford, UK), and rabbit anti-DSpd-2 polyclonal (1:500; Dix and Raff, 2007). Secondary antibodies were from Thermo Fisher Scientific: Goat anti-Mouse IgG (H+L) Cross-Adsorbed Secondary Antibody, Alexa Fluor 488 (A11001), Goat anti-Rabbit IgG (H+L) Cross-Adsorbed Secondary Antibody, Alexa Fluor 488 (A-11008), Goat anti-Mouse IgG (H+L) Cross-Adsorbed Secondary Antibody, Alexa Fluor 568 (A-11004), Goat anti-Rabbit IgG (H+L) Cross-Adsorbed Secondary Antibody, Alexa Fluor 568 (A-11011), Goat anti-Guinea Pig IgG (H+L) Highly Cross-Adsorbed Secondary Antibody, Alexa Fluor 633 (A-21105). Hoechst 33342 (H1399; Thermo Fisher Scientific) was used to stain DNA.

For Western blotting, we used mouse anti-γ-tubulin monoclonal (1:250; GTU88; Sigma-Aldrich), rabbit anti-MBP polyclonal (1:1,000; gift from Jordan Raff), rabbit anti-Cnn (N-term) polyclonal (1:1,000; gift from Jordan Raff), sheep anti-Cnn (C-term) polyclonal (1:1,000; gift from Jordan Raff), rabbit anti-Spd-2 polyclonal (1:500; gift from Jordan Raff), rabbit anti-Grip71 polyclonal (1:250; #2005268; CRB), and mouse anti-Actin monoclonal (1:1,000; gift from Jordan Raff).

### Fixed brain analysis
For the analysis of centrosomal fluorescence levels of γ-tubulin or other PCM components, third instar larval brains were dissected and incubated in 100 μM colchicine in Schneider's medium for 1 h at 25°C to depolymerize microtubules. This prevents centrosomes in *cnn* mutants from "rocketing" and transiently losing their PCM (Lucas and Raff, 2007), allowing a more accurate quantification of PCM recruitment (Conduit et al., 2014). Brains were fixed in 4% paraformaldehyde containing 100 mM PIPES, 1 mM MgSO$_4$, and 2 mM EGTA pH 6.95 for 20 min at room temperature, washed in PBS 3 × 5 min, and then incubated in 45% acetic acid for 30 s and then 60% acetic acid for 3 min. The brains were then squashed under a coverslip using a pencil to hit down on the coverslip (with blotting paper protecting the coverslip) and then plunged into liquid nitrogen. The coverslips were rapidly removed using a razor blade and the slides with attached brain material were incubated in methanol at –20°C for 8 min. The slides were then washed in PBT 3 × 20 min, air dried, and then the appropriate primary antibody solution was added within the boundary of a hydrophobic PAP pen line and the slides were incubated in a humid chamber overnight at 4°C. Slides were then washed 3 × 20 min in PBT, air dried, and then the appropriate secondary antibody solution was added within the boundary of the PAP pen line and the slides were incubated in a humid chamber for 3 h at 21°C. Slides were then washed 3 × 20 min in PBT, air dried, and mounted by adding 10 μl NPG-Glycerol mounting medium (2% n-propyl-gallate (MERCK 02370), 49% PBS, 49% glycerol pH = 7.4).

Images were collected on either a Leica SP5 point scanning upright confocal system run by LAS AF software using a 63 × 1.3NA glycerol objective (1156194; Leica) or a Zeiss Axio Observer.Z1 inverted CSU-X1 Yokogowa spinning disk system with two ORCA Fusion camera (Hamamatsu) run by Zeiss Zen2 acquisition software using a 60 × 1.4NA oil immersion lens (Zeiss), or a Zeiss LSM700 upright confocal microscope run by Zeiss Zen acquisition software using a 63 × 1.3NA oil immersion lens (Zeiss). All images were collected at "room temperature," ~21°C. See below for details on image processing and analysis.

At least five images containing multiple cells in both mitosis (as shown by positive Phospho-Histone H3 staining) and interphase were collected for each brain. Each data point on a graph represents the average signal from all the centrosomes quantified in a single brain. Typically, between 30 and 50 centrosomes were analyzed per cell cycle stage (interphase or mitosis) per brain.

For assessing the ability of centrosomes to organize microtubules during prophase, third instar larval brains were treated and imaged as above, except that the colchicine incubation step was omitted. A prophase cell was scored as positive when at least

one centrosome had an associated α-tubulin signal. For measuring the distance of centrosomes from the spindle pole during prometaphase, measurements were made between the center of the Asl signal (centrosome) and the spindle pole (center of the α-tubulin signal at the spindle pole).

## Fixed microtubule re-growth assay

Third instar larval brains of the appropriate genotype were dissected and incubated on ice in Schneider's medium for 40 min. Empirical tests showed that a 40-min incubation was necessary to efficiently depolymerize centrosomal microtubules. Larval brains were then either rapidly fixed on ice in 16% paraformaldehyde containing 100 mM PIPES, 1 mM $MgSO_4$, and 2 mM EGTA pH 6.95 for 5 min (T0 brains), or were quickly transferred to room temperature for 30 s and then rapidly fixed in 16% paraformaldehyde containing 100 mM PIPES, 1 mM $MgSO_4$, and 2 mM EGTA pH 6.95 for 5 min at room temperature. Subsequently, the brains were processed as above. Images were collected on the Leica SP5 point scanning upright confocal system run by LAS AF software using a 63 × 1.3NA glycerol objective (1156194; Leica) at room temperature, ~21°C. See below for details on image processing and analysis.

## Live analysis of microtubule and EB1-GFP comets during cooling warming cycles

A CherryTemp device from CherryBiotech was used to modulate the temperature of larval brain cells. Third instar larval brains were dissected and semi-squashed between a coverslip and the CherryTemp thermalization chip in Schneider's medium. The *cnn,grip71,grip163* mutant samples and their respective controls were imaged on a Leica DM IL LED inverted microscope controlled by μManager software and coupled to a RetigaR1 monochrome camera (QImaging) and a CoolLED pE-300 Ultra light source using a 63 × 1.3NA oil objective (11506384; Leica). The *cnn^{ΔCM1},grip71,grip163* mutant samples and their respective controls were imaged on a Leica DMi8 inverted microscope controlled by μManager software and coupled to a BSI Prime Express monochrome camera (QImaging) and a CoolLED pE-300 Ultra light source using a 63 × 1.3NA oil objective (11506384; Leica). The temperature was changed from 20°C to 5°C and back to 20°C for microtubule depolymerization and repolymerization, respectively. Temperature changes induce movements in the glass and the focus was manually adjusted to keep as many frames in focus as possible during the temperature shifts. For Jupiter-mCherry Videos, Z-stacks with gaps of 500 nm were acquired every 6 s; for EB1-GFP Videos, Z-stacks with gaps of 300 nm were acquired every second. For the quantification of Jupiter-mCherry in the *cnn,grip71,grip163* mutant experiment, 12 and 10 centrosomes from 7 and 10 control and *cnn,grip71,grip163* mutant cells were analyzed, respectively. For the quantification of Jupiter-mCherry in the *cnn^{ΔCM1},grip71,grip163* mutant experiment, 12 and 11 centrosomes from 8 and 9 control and *cnn^{ΔCM1},grip71,grip163* mutant cells were analyzed, respectively. GraphPad Prism was used to generate the one-phase exponential decay models and exponential plateau models that are fitted to the depolymerisation and nucleation/regrowth phases, respectively.

## Image analysis and statistics

All images were processed in Fiji (ImageJ). Each Z-stack image was reconstructed by maximum intensity Z-axis projection. PCM or microtubule levels at centrosomes were calculated by measuring the total fluorescence in a boxed or circular region around the centrosome and subtracting the local cytoplasmic background fluorescence. GraphPad Prism was used for statistical analysis. When parametric tests were used, tests for normality were first performed using D'agostino and Pearson tests, Anderson–Darling tests, Shapiro–Wilk tests, and Kolmogorov–Smirnov tests. Datasets were considered to be normally distributed when at least one test passed the normality test. All *t* tests were two-sided. When using one-way ANOVA, we assumed equal SDs. We corrected for multiple comparisons using Šídák hypothesis testing and multiplicity corrected P values for each comparison were reported. When using Kruskal–Wallis tests, we corrected for multiple comparisons using Dunn's hypothesis testing and multiplicity corrected P values for each comparison were reported. For fitting and comparing the centrosomal Jupiter-mCherry signals during cooling and warming, one-phase exponential decay models and "exponential plateau" models generated in GraphPad Prism using least squares fit were fitted to the depolymerization and nucleation/regrowth phases, respectively. To log transform the data in Fig. 5 H, we took the $log_{10}$ values. The following tests were used to make comparisons between datasets: For Fig. 1 B, we used a one-way ANOVA with correction for multiple comparisons; for Fig. 1 D, Fig. 2, B and D; Fig. 4 B; and Fig. S1 F we used paired *t* tests; for Fig. 3, G and H; and Fig. 4, F and G the fits were compared using an extra sum-of-squares *F* test; for Fig. 5 F we used one-way Chi-squared tests; for Fig. 5, G and H we used Kruskal–Wallis tests with correction for multiple comparisons using Dunn's. Details of *N* numbers, the statistical tests, and models used can be found in the figure legends.

## Recombinant protein cloning, expression, and purification

The Cnn-C-N and Cnn-C-N^{T27E,S186D} fragments comprise amino acids 1–255 of Cnn-C and were generated previously (Tovey et al., 2021). Briefly, for the Cnn-C-N fragment, the region encoding aa1–255 of Cnn was inserted into a pDONR vector and then a pDEST-HisMBP destination vector (#11085; Addgene) by Gateway cloning (Thermo Fisher Scientific). For the Cnn-C-N^{T27E,S186D} fragment, the pDONR-Cnn-C-N entry clone was linearized by digestion, omitting the phospho-residues to be replaced, and a fragment generated by GENEWIZ that contained the T27E and S186 mutations and appropriate overlapping ends was cloned in using HiFi Assembly (NEB). The region of interest was then cloned into a pDEST-HisMBP destination vector via Gateway cloning. We generated the Cnn-C-N^{T27E,R101Q,E102A,F115A,S186D} fragment in a similar manner. Briefly, the pDEST-HisMBP (#11085; Addgene) vector containing aa1-255 of Cnn was digested with KpnI and SspI, and a complementary fragment containing the point mutations was cloned into the cut vector using HiFi technology (NEB). The complementary fragment was generated by GENEWIZ. For the Cnn-C-N^{T27E,F115A,S186D} and Cnn-C-N^{T27E,R101Q,E102A,S186D} fragments we used QuikChange (Agilent) to introduce the F115A or R101Q,E102A

mutations into a pDONR vector containing the T27E,S186D mutations that had been generated previously (Tovey et al., 2021), and then used Gateway (Thermo Fisher Scientific) to clone the Cnn sequences into a pDEST-MBP-PreSci-His-Strep vector. The pDEST-MBP-PreSci-His-Strep vector was made by modifying a pDEST-His-MBP-PreSci-His vector by amplifying three separate sections of the vector by PCR and using overlapping primers in a "non-templated" PCR reaction and then combining the four fragments using HiFi assembly. The final vectors were transformed into BL21-DE3 cells and proteins were purified using gravity flow amylose resin (New England Biolabs) affinity chromatography. Peak elution fractions were diluted 1:1 with glycerol and stored at –20°C.

## Immunoprecipitation

Immunoprecipitation was carried out as follows: 1 g/ml of wild-type embryos were homogenized in a homogenization buffer containing 50 mM HEPES, pH 7.6, 1 mM MgCl₂, 1 mM EGTA, 50 mM KCl supplemented with PMSF 1:100, Protease Inhibitor Cocktail (1:100, Sigma-Aldrich) and DTT (1 M, 1:1,000). Extracts were clarified by centrifugation twice for 15 min at 16,000 rcf at 4°C and 100 µl embryo extract was rotated at 4°C overnight with 30 µl magnetic ProteinA dynabeads (Life Technologies) coupled with anti-MBP antibodies (gift from Jordan Raff) and MBP-Cnn fragments. Beads were washed five times for 1 min each in PBS + 0.1% triton (PBST), boiled in 50 µl 2× sample buffer (BioRad), and separated from the eluted IP sample using a magnet.

## Gel electrophoresis and Western blotting

Samples were run on 4–20% TGX Precast Gels (BioRad), alongside 5 µl Precision Plus WesternC Standard markers (BioRad). For the Western blots in Fig. 1 and Fig. S1, semi-dry Western blotting was carried out using TransBlot Turbo 0.2 µm nitrocellulose membrane transfer packs (BioRad) and a TransBlot Turbo transfer system running at 1.3A, up to 25 V, for 7 min. For the Western blots in Fig. S6, wet transfer was performed in 25 mM Tris, 192 mM glycine, pH 8.3, 20% methanol (vol/vol), and 0.1% SDS—the Western blot was run at a constant 50 mA overnight. Membranes were stained with Ponceau and washed, first with distilled water then with milk solution (PSBT + 4% milk powder), and then blocked in milk solution for 1 h at room temperature. Sections of blots were incubated with primary antibodies as indicated in the figures. Blots were incubated with species-appropriate horseradish peroxidase (HRP)-conjugated secondary antibodies (1:2,000 in PSBT + 4% milk powder, ImmunoReagents) for 45 min at room temperature, washed in PSBT three times for 15 min each, and then incubated with ECL substrate (BioRad ECL Clarity or Thermo Fisher Scientific SuperSignal West Femto Max) for 2 min.

For Western blotting of brain samples to examine the expression of the cnn^ΔCM1 allele, we dissected eight brains for each genotype and boiled them at 95°C for 10 min in 30 µl 4× sample buffer. 10 µl was loaded onto a 4–20% gel (BioRad) and ran for 50 min at 200 V. Proteins were transferred onto nitrocellulose membrane using the semi-dry TransTurbo Blot (BioRad) and incubated with the appropriate primary and then secondary antibodies.

Western blot images in Fig. 1 and Fig. S1 were generated using a BioRad ChemiDoc; Western blot images in Fig. S6 were generated by scanning x-ray films generated by a Western blot film developer machine.

## Online supplemental material

Fig. S1 shows the characterization of the Cnn^ΔCM1 allele, including sequencing results and expression levels. Fig. S2 shows the fluorescent profiles of individual centrosomes in control or cnn,grip71,g163 mutant cells during cooling/warming cycles. Fig. S3 shows seven examples of how alpha-tubulin does not concentrate at centrosomes after cold-induced microtubule depolymerization. Fig. S4 shows that the distribution of cells with and without centrosomes and with and without reformed spindles is similar in cnn,grip71,grip163 mutant and cnn,grip71,-grip163,msps mutant cells after 30 s of warming post cooling. Fig. S5 shows the results of a qualitative "spindle quality" analysis for wild-type, cnn,grip71,g163 mutant, and cnn,grip71,g163,msps mutant cells. Fig. S6 shows results of Western blots revealing the absence of Cnn or Spd-2 proteins in cnn and spd-2 mutant brain samples. Videos 1 and 2 show the Jupiter-mCherry signal within a control and cnn,grip71,grip163 mutant cell, respectively, as the cells go through cooling/warming cycles. Videos 3 and 4 show the EB1-GFP signal within a control and cnn,grip71,grip163 mutant cell, respectively, as the cells go through cooling/warming cycles.

## Data availability

The Excel sheets and image files containing the raw data underlying all figures are available from the corresponding author upon reasonable request.

## Acknowledgments

We thank Jordan Raff for sharing antibodies and fly lines. We thank Roger Karess for his invaluable input and critical reading of the manuscript. The work benefited from the Imaging Facility, Department of Zoology, University of Cambridge, supported by Matt Wayland and a Sir Isaac Newton Trust Research Grant (18.07ii(c)), and from the ImagoSeine at the IJM, Paris. For the purpose of Open Access, the author has applied a CC BY public copyright license to any Author-Accepted Manuscript version arising from this submission.

This research was supported by the Centre National de la Recherche Scientifique CNRS, Université Paris Cité, a Wellcome Trust and Royal Society Sir Henry Dale fellowship awarded to P.T. Conduit [105653/Z/14/Z], by a Chaire d'excellence grant from the IdEx Université Paris Cité ANR-18-IDEX-0001 awarded to P.T. Conduit, by an ATIP Avenir award funded by the Fondation Bettencourt Schueller awarded to P.T. Conduit, and by an Association pour la Recherche sur le Cancer grant (PJA 20181208148) awarded to A. Guichet.

Author contributions: Z. Zhu produced fly stocks by combining alleles, performed cell imaging experiments, in particular establishing and executing the cooling-warming assays, and analyzed data. I. Becam contributed to the revision of the manuscript by generating fly lines, performing cell imaging

experiments, analyzing data, and providing feedback on the manuscript. C.A. Tovey and A. Elfarkouchi performed the immunoprecipitation experiments, including cloning and protein purification of the Cnn fragments, and C.A. Tovey provided feedback on the manuscript. E.C. Yen contributed to establishing the fixed cell cooling-warming experiment. A. Guichet obtained funding to generate the EB1-GFP fly line via InDroso with coordination from F. Bernard. P.T. Conduit obtained funding, designed the study, generated fly stocks, performed cell imaging experiments, analyzed data, and wrote the manuscript.

Disclosures: The authors declare no competing interests exist.

Submitted: 9 December 2022

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

# Supplemental material

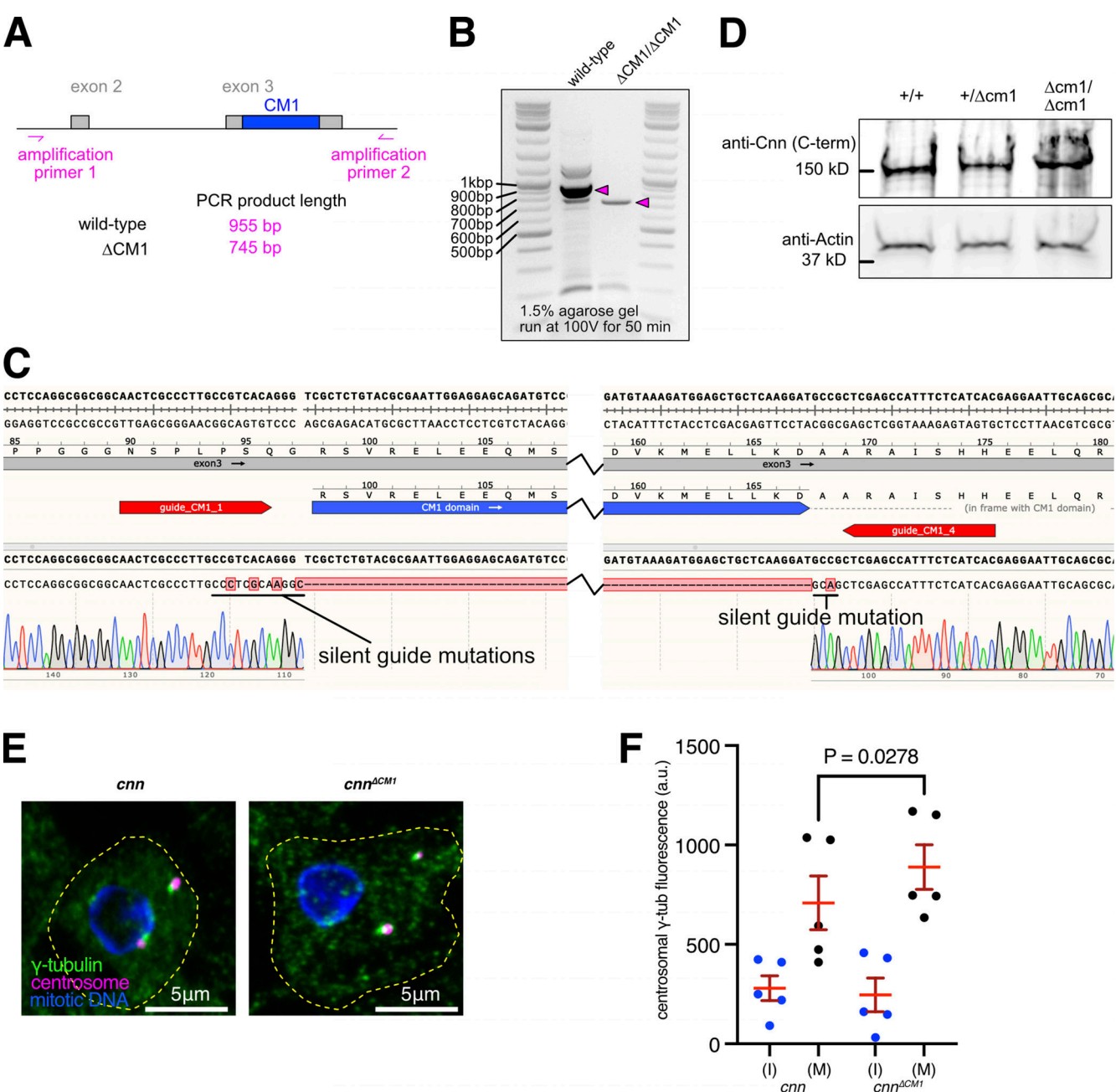

Figure S1. **Centrosomes from cnn^{ΔCM1} mutants accumulate slightly more γ-tubulin than centrosomes from cnn null mutants. (A)** Cartoon representation of region amplified to check for CM1 deletion mutants. **(B)** Gel showing DNA bands from PCR reactions when using the primers shown in A on either wild-type flies or CM1 deletion flies, as indicated. **(C)** Excerpt from SnapGene showing the sequencing result when using amplification primer 2 to sequence the PCR product generated using amplification primer 1 and amplification primer 2 on CM1 deletion flies. The position of the guide RNAs used when making the deletion is indicated. Note that silent mutations were introduced to prevent Cas9 from recutting the DNA after the recombination event. **(D)** Western blot of larval brain extracts from different genotypes, as indicated, probed with C-terminal anti-Cnn polyclonal antibodies. Note that the ~8 kD difference in size between wild-type Cnn, which runs at ~150 kD, and Cnn^{ΔCM1} is not discernible. **(E)** Fluorescence images of mitotic *Drosophila* brain cells from either *cnn* or *cnn^{ΔCM1}* mutant third instar larvae immunostained for γ-tubulin (green), mitotic DNA (blue), and Asl (centrioles, magenta). Note how the γ-tubulin signal in *cnn^{ΔCM1}* mutant cells is also offset from the Asl signal, indicating that removing the CM1 domain affects the Cnn's ability to form a proper centrosomal scaffold. Scale bars are 5 μm. **(F)** Graph showing average fluorescence intensities of interphase (blue dots) and mitotic (black dots) centrosomes from either *cnn* or *cnn^{ΔCM1}* mutant brains (as indicated below). Each datapoint represents the average centrosome value from one brain. *N* = 5 for each dataset. Mean and SEM are indicated. Brains from the different genotypes were paired on slides (one slide per pair) allowing a two-sided paired *t* test to compare the mean values between mitotic centrosomes. Note that γ-tubulin accumulation at mitotic centrosomes is only slightly higher in *cnn^{ΔCM1}* mutant cells, indicating that either the Cnn-dependent pool of Spd-2 is not an efficient recruiter of γ-TuRCs or that recruitment of the Cnn-dependent pool of Spd-2 is perturbed in *cnn^{ΔCM1}* mutant cells or both. Source data are available for this figure: SourceData FS1.

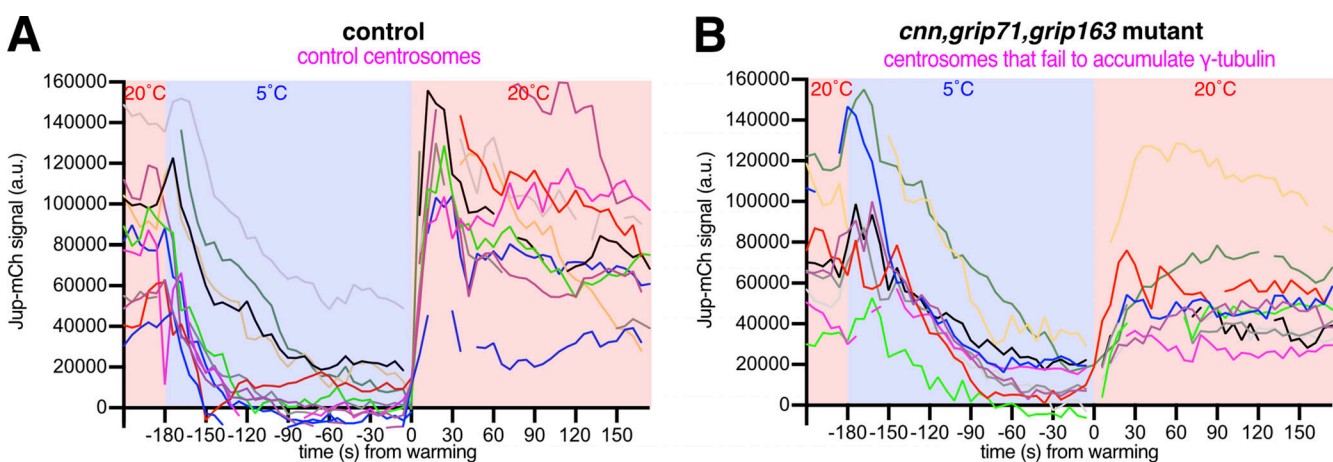

Figure S2. **Plots of individual centrosome Jup-mCherry values during cooling–warming experiments. (A and B)** Graphs plotting the centrosomal signal (after subtraction of cytosolic signal) of Jupiter-mCherry within living *Drosophila* control (A) and *cnn,grip71,grip163* mutant (B) third instar larval brain cells as they were cooled to 5°C for ~3 min and then rapidly warmed to 20°C. Time in seconds relative to the initiation of warming (0 s) is indicated. Note how the centrosomal Jupiter-mCherry signal does not always reach cytosolic levels (i.e., 0), indicating that microtubules were not fully depolymerized from all centrosomes, but note also that even when the Jupiter-mCherry signal did reach cytosolic levels there was still a rapid increase after warming, indicating that the increase in signal after warming is not simply due to regrowth of partially depolymerized microtubules.

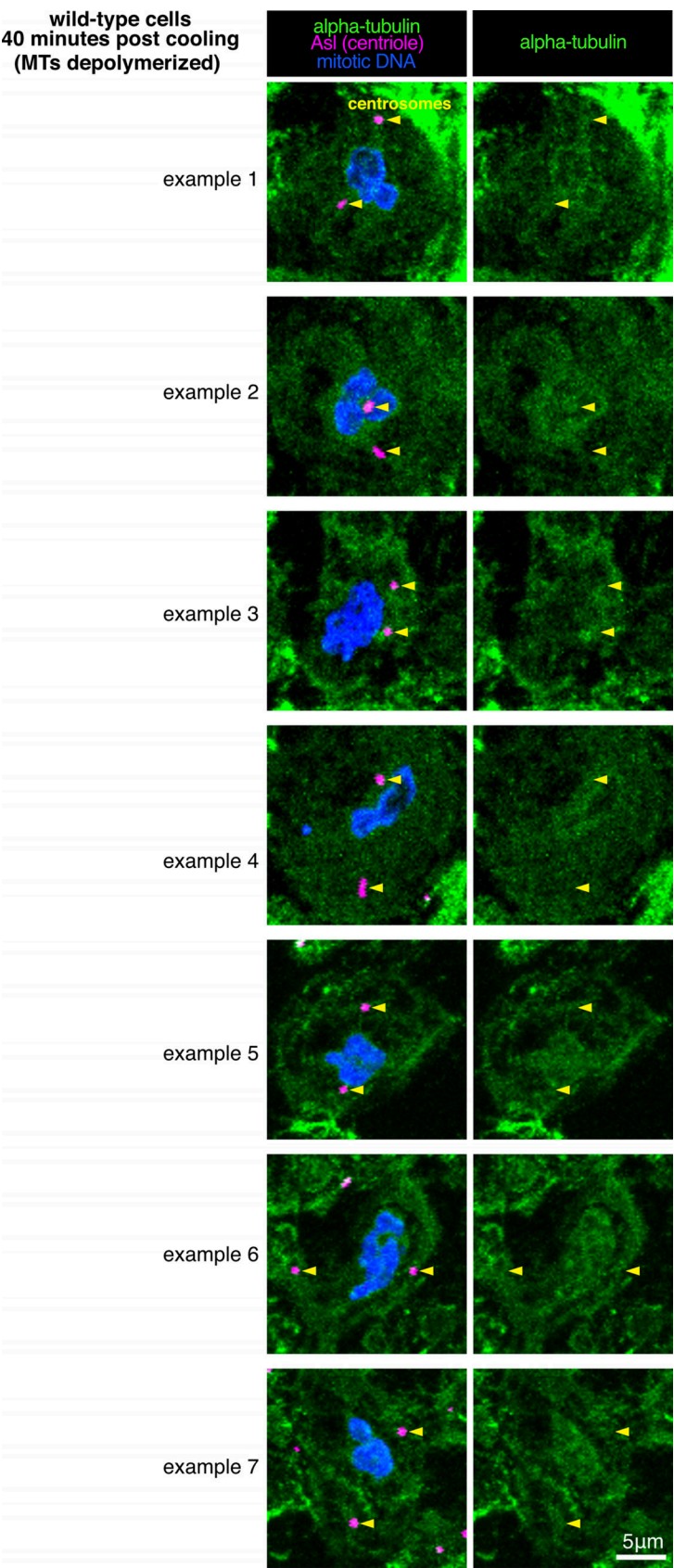

Figure S3. ***Drosophila* centrosomes do not concentrate α/β-tubulin.** Fluorescence images of mitotic *Drosophila* brain cells from wild-type third instar larval brains that had been cooled for 40 min on ice immunostained for alpha-tubulin (microtubules, green), mitotic DNA (blue), and Asl (centrioles, magenta). Note how there is no tubulin signal at centrosomes that is above cytosolic levels. Note also that example 1 (top panel) is the same cell as shown in Fig. 3 B, top panel.

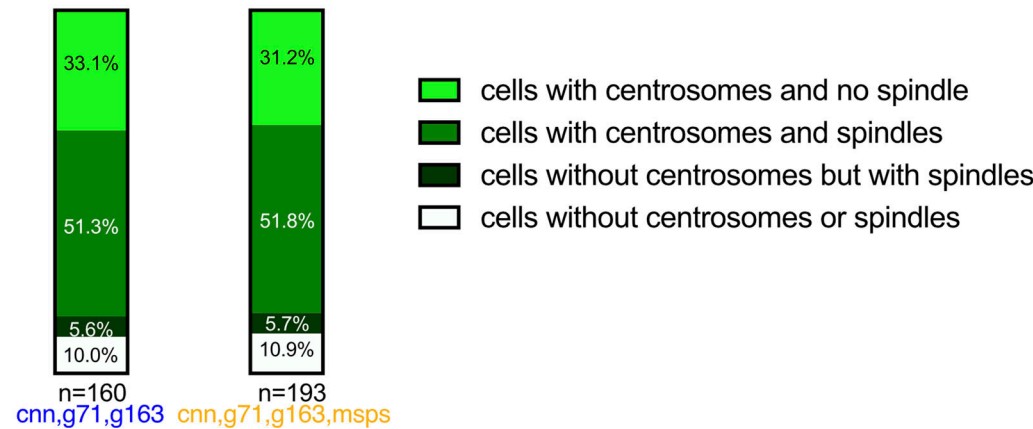

Figure S4. **The distribution of cells with and without centrosomes and with and without re-formed spindles is similar in *cnn,grip71,grip163* mutant and *cnn,grip71,grip163,msps* mutant cells after 30 s of warming after cooling.** Parts of a whole graph show the proportion of cells that either contain centrosomes or do not, and that have either formed a spindle or have not, in cells fixed and immunostained for alpha-tubulin (microtubules, green), mitotic DNA (blue), and Asl (centrioles, magenta) after 30 s of warming post cooling from either *cnn,grip71,grip163* or *cnn,grip71,grip163,msps* mutants, as indicated. *N* numbers are indicated, and each N represents a single cell.

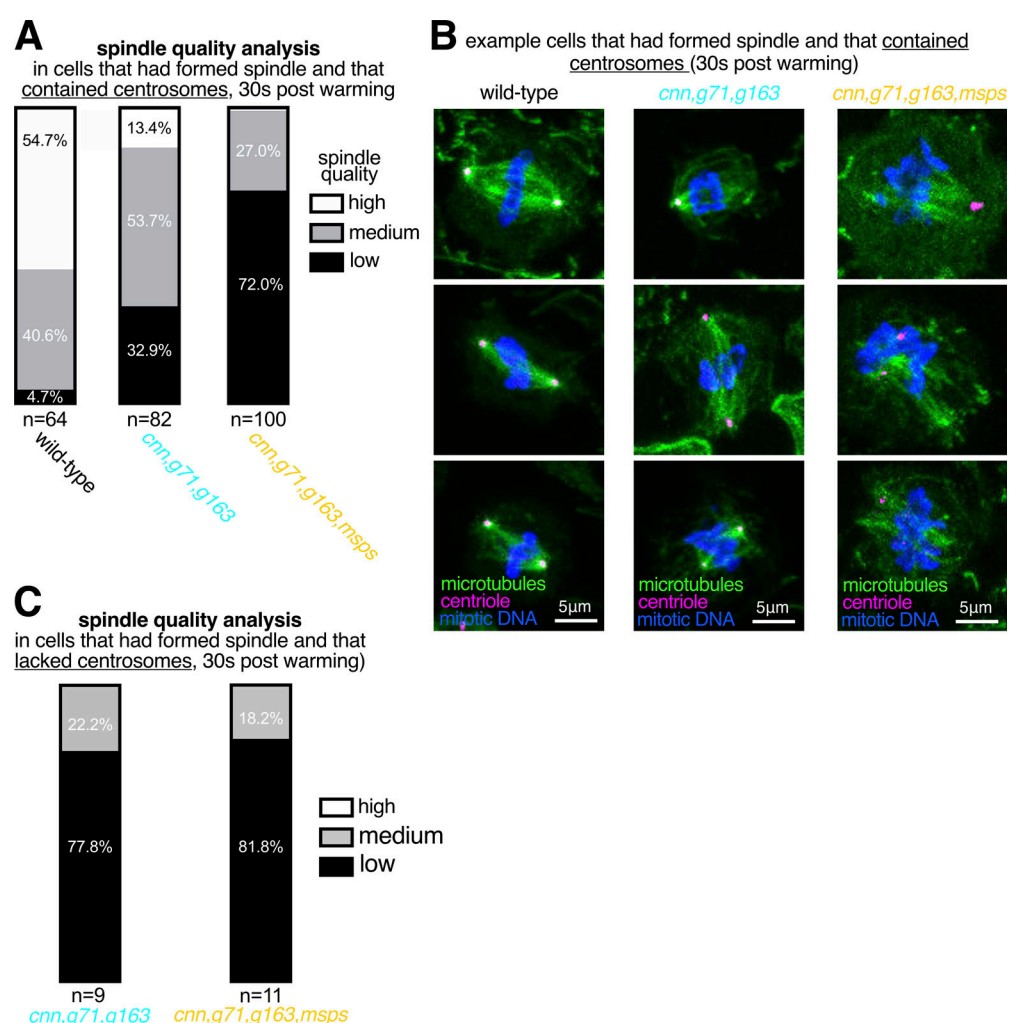

Figure S5. **Spindles form less robustly in cells depleted of Msps in addition to Cnn, Grip71, and Grip163[GCP6]. (A–C)** Parts of a whole graphs (A and C) and images (B) represent analyses of mitotic spindle quality in cells fixed and immunostained for alpha-tubulin (microtubules, green), mitotic DNA (blue), and Asl (centrioles, magenta) after 30 s of warming post cooling from either wild-type, *cnn,grip71,grip163*, or *cnn,grip71,grip163,msps* mutants, as indicated. *N* numbers are indicated, and each *N* represents a single cell. Only cells that had centrosomes and that had already formed a spindle were analyzed in A and B, and only cells lacking centrosomes but that had formed a spindle were analyzed in C. Note that wild-type cells always contained centrosomes and so were not analyzed in C. Note also how spindle quality is frequently low in *cnn,grip71,grip163,msps* mutants (A and B), but the difference in spindle quality between mutant types is not apparent in cells lacking centrosomes (C). Note also that some cells lacking Cnn have abnormal numbers of centrosomes due to centrosome segregation problems during cell division (Conduit et al., 2010).

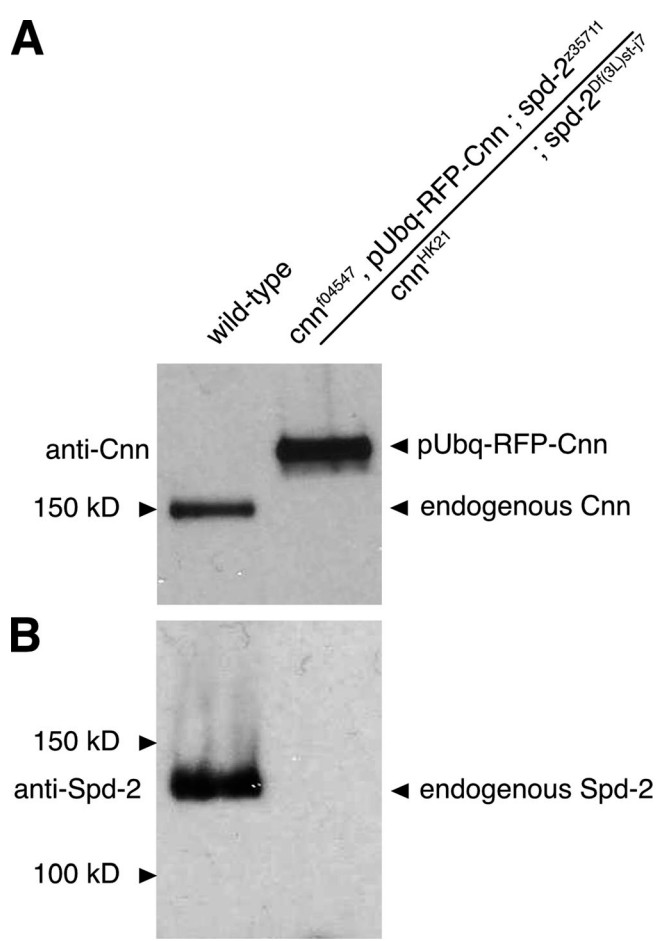

Figure S6. **Brains from flies carrying mutant alleles for *cnn* or *spd-2* display no observable Cnn or Spd-2 protein on Western blots. (A and B)** Western blots of larval brain samples from wild-type and cnn,spd-2,pUbq-RFP-Cnn flies probed with anti-Cnn (A) and anti-Spd-2 (B). Note how anti-Cnn recognizes endogenous Cnn in the wild-type sample and the larger exogenous pUbq-RFP-Cnn in the cnn,spd-2,pUbq-RFP-Cnn mutant sample, but that no endogenous Cnn is detected in the cnn,spd-2,pUbq-RFP-Cnn mutant sample. Note also how anti-Spd-2 recognizes endogenous Spd-2 in the wild-type sample but not in the cnn,spd-2,pUbq-RFP-Cnn mutant sample. Source data are available for this figure: SourceData FS6.

Video 1. **Microtubule depolymerization and regrowth at centrosomes in control cells.** Video showing the Jupiter-mCherry signal (marking microtubules) within a living control cell during a cooling–warming experiment. Cooling to 5°C begins at −174 s and warming to 20°C begins at 0 s. Note how the Jupiter-mCherry signal decreases gradually during cooling and then recovers immediately at the two centrosomes during warming.

Video 2. **Microtubule depolymerization and regrowth at centrosomes in *cnn,grip71,grip163* mutant cells.** Video showing the Jupiter-mCherry signal (marking microtubules) within a living *cnn,grip71,grip163* mutant cell during a cooling–warming experiment. Cooling to 5°C begins at −186 s and warming to 20°C begins at 0 s. Only one centrosome is present (spindle pole on the left). Note how the Jupiter-mCherry signal decreases gradually during cooling and then recovers immediately at the centrosome during warming, but that this recovery is slower and less intense than in control cells.

Video 3. **Behavior of EB1-GFP comets at centrosomes during a cooling–warming cycle in control cells.** Video showing the EB1-GFP signal (marking growing microtubule ends) within a living control cell during a cooling–warming experiment. Cooling to 5°C begins at −54 s and warming to 20°C begins at 0 s. Note how the EB1-GFP signal disappears immediately on cooling and then dramatically reappears and spreads outward from the two centrosomes during warming.

Video 4.   **Behavior of EB1-GFP comets at centrosomes during a cooling-warming cycle in *cnn,grip71,grip163* mutant cells.** Video showing the EB1-GFP signal (marking growing microtubule ends) within a living *cnn,grip71,grip163* mutant cell during a cooling–warming experiment. Cooling to 5°C begins at −276 s and warming to 20°C begins at 0 s. Note how the EB1-GFP signal does not disappear immediately on cooling, unlike in control cells. The signal does disappear fully prior to warming and then reappears from the centrosomes and chromosomal regions during warming, spreading outward. Note also that the centrosomes do not remain in focus throughout the movie. The centrosome at the spindle pole in the lower half of the cell is in focus throughout most of the movie, but this centrosome is out of focus for ~30 s after warming due to fluctuations in the cover glass during the temperature change.

