## [Peer Review File · The Journal of Cell Biology]

Multifaceted modes of γ -tubulin complex recruitment and microtubule nucleation at mitotic centrosome

Zihan Zhu, Isabelle Becam, Corinne Tovey, Abir Elfarkouchi, Eugenie Yen, Fred BERNARD, Antoine Guichet, and Paul Conduit

Corresponding Author(s): Paul Conduit, Institut Jacques Monod

Review Timeline:

Submission Date:	2022-12-09
Editorial Decision:	2023-01-03
Revision Received:	2023-05-18
Editorial Decision:	2023-06-28
Revision Received:	2023-07-05
Editorial Decision:	2023-07-18
Revision Received:	2023-07-20

Monitoring Editor: Arshad Desai

Scientific Editor: Tim Spencer

Transaction Report:

DOI: <https://doi.org/10.1083/jcb.202212043>

January 3, 2023

Re: JCB manuscript #202212043T

Dr. Paul T Conduit
Institut Jacques Monod
15 rue H el ene Brion
Paris 75205
France

Dear Dr. Conduit,

Thank you for submitting your manuscript entitled "Multifaceted modes of γ -tubulin complex recruitment and microtubule nucleation at mitotic centrosomes". We have discussed your manuscript, reviews and revision plan and have also sought input from one of the original Review Commons reviewers. We apologize for the delay in providing you with a decision but we felt it important to get input on your proposed plan. While we agree with the reviewers that your study has the potential to provide an important conceptual advance with respect to the different pathways by which centrosomes nucleate microtubules, there are also significant concerns that unfortunately preclude publication of the current version of the manuscript in JCB.

We recognize that you have addressed most of the reviewers' points that required textual/figure changes or providing additional information, for example about the specific alleles being used. However, for a successful revision, it would be essential that you address the following points of reviewers #1 and #3 with new experimentation. While we appreciate the amount of time and effort that would need to be put into generating new fly lines and collecting data, we would require that you address point 2 raised by rev #1 -as proposed in your revision plan, by testing whether γ -tubulin can be recruited to mitotic centrosomes in Cnn Δ CM1,grip71,grip163 flies, which have a PCM matrix, and then should no γ -tubulin be recruited, by analyzing microtubule nucleation in live cells under this condition in Cnn Δ CM1, grip71, grip163, Jup-mCherry flies. In addition, we would require that you address the first two points raised by rev #3 accordingly as proposed again in your revision plan. You would need to add a cnn mutant-only control to Fig. 2B (by adding a Cnn vs. Cnn,grip71 comparison), and to compare γ -tubulin recruitment in a Cnn-null vs. Cnn Δ CM1 background.

Note that a substantial amount of additional experimental data likely would be needed to satisfactorily address the concerns of the reviewers. The typical timeframe for revisions is three to four months. While most universities and institutes have reopened labs and allowed researchers to begin working at nearly pre-pandemic levels, we at JCB realize that the lingering effects of the COVID-19 pandemic may still be impacting some aspects of your work, including the acquisition of equipment and reagents. Therefore, if you anticipate any difficulties in meeting this aforementioned revision time limit, please contact us and we can work with you to find an appropriate time frame for resubmission. Please note that papers are generally considered through only one revision cycle, so any revised manuscript will likely be either accepted or rejected.

If you choose to revise and resubmit your manuscript, please also attend to the following editorial points. Please direct any editorial questions to the journal office.

GENERAL GUIDELINES:

Text limits: Character count is < 40,000, not including spaces. Count includes title page, abstract, introduction, results, discussion, and acknowledgments. Count does not include materials and methods, figure legends, references, tables, or supplemental legends.

Figures: Your manuscript may have up to 10 main text figures. To avoid delays in production, figures must be prepared according to the policies outlined in our Instructions to Authors, under Data Presentation, <https://jcb.rupress.org/site/misc/ifora.xhtml>. All figures in accepted manuscripts will be screened prior to publication.

IMPORTANT: It is JCB policy that if requested, original data images must be made available. Failure to provide original images upon request will result in unavoidable delays in publication. Please ensure that you have access to all original microscopy and blot data images before submitting your revision.

Supplemental information: There are strict limits on the allowable amount of supplemental data. Your manuscript may have up to 5 supplemental figures. Up to 10 supplemental videos or flash animations are allowed. A summary of all supplemental material should appear at the end of the Materials and methods section.

Please note that JCB now requires authors to submit Source Data used to generate figures containing gels and Western blots with all revised manuscripts. This Source Data consists of fully uncropped and unprocessed images for each gel/blot displayed in the main and supplemental figures. Since your paper includes cropped gel and/or blot images, please be sure to provide one

Source Data file for each figure that contains gels and/or blots along with your revised manuscript files. File names for Source Data figures should be alphanumeric without any spaces or special characters (i.e., SourceDataF#, where F# refers to the associated main figure number or SourceDataFS# for those associated with Supplementary figures). The lanes of the gels/blots should be labeled as they are in the associated figure, the place where cropping was applied should be marked (with a box), and molecular weight/size standards should be labeled wherever possible.

If you choose to resubmit, please include a cover letter addressing the reviewers' comments point by point. Please also highlight all changes in the text of the manuscript.

Regardless of how you choose to proceed, we hope that the comments below will prove constructive as your work progresses. We would be happy to discuss them further once you've had a chance to consider the points raised. You can contact the journal office with any questions, cellbio@rockefeller.edu or call (212) 327-8588.

Thank you for thinking of JCB as an appropriate place to publish your work.

Sincerely,

Arshad Desai
Monitoring Editor
Journal of Cell Biology

Lucia Morgado-Palacin, PhD
Scientific Editor
Journal of Cell Biology

Response to Reviewer comments – RC-2022-01689

We thank all the Reviewers for their highly constructive reviews. Below, I document our response to each point. Please note that most of the revisions have already been seen by the Editors and one of the original reviewers, who agreed on a final revision plan back in January (see text in green below). Nevertheless, I have been asked to include all of the revision responses in this document for the benefit of the other two Reviewers. I have therefore colour coded the responses – responses in red are the original responses we sent to JCB after receiving the reviews from Review Commons – these were agreed to be satisfactory by the Editors and one of the reviewers. Responses in blue are the new revisions we have carried out since the initial response from JCB in January.

Email from Editors 3rd January 2023:

*"We recognize that you have addressed most of the reviewers' points that required textual/figure changes or providing additional information, for example about the specific alleles being used. However, for a successful revision, it would be essential that you address the following points of reviewers #1 and #3 with new experimentation. While we appreciate the amount of time and effort that would need to be put into generating new fly lines and collecting data, we would require that you address point 2 raised by rev #1 -as proposed in your revision plan, by testing whether γ -tubulin can be recruited to mitotic centrosomes in *Cnn Δ CM1,grip71,grip163* flies, which have a PCM matrix, and then should no γ -tubulin be recruited, by analyzing microtubule nucleation in live cells under this condition in *Cnn Δ CM1, grip71, grip163, Jup-mCherry* flies. In addition, we would require that you address the first two points raised by rev #3 accordingly as proposed again in your revision plan. You would need to add a *cnn* mutant-only control to Fig. 2B (by adding a *Cnn* vs. *Cnn,grip71* comparison), and to compare γ -tubulin recruitment in a *Cnn*-null vs. *Cnn Δ CM1* background."*

Reviewer #1 (Evidence, reproducibility and clarity (Required)):

In their study, Zhu and colleagues study how the centrosome proteins Spd-2 and Cnn in *Drosophila* recruit gamma-tubulin complexes to centrosomes, which is an important step in mitotic spindle formation. The authors make use of mutant flies and RNAi and find that the two factors Spd-2 and Cnn together are responsible for mitotic centrosomal accumulation of gamma-tubulin. By inactivating Spd-2 or Cnn separately, the authors show that Cnn appears to recruit the large share of mitotic gamma-tubulin pool by its CM1-domain. Interestingly, this involves only gamma-TuSCs (subcomplexes of gamma-TuRC) and not gamma-TuRCs. A smaller pool is recruited by Spd-2, and this pool depends on gamma-tubulin complex proteins that are only present in pre-assembled, complete gamma-TuRCs. This suggests that *Drosophila* makes microtubule nucleation templates in two ways. First, as in yeast, by direct recruitment of gamma-TuSCs to mitotic centrosomes, where additionally oligomerization needs to happen. And second, by recruitment and activation of preassembled gamma-TuRCs. Inactivation of both Cnn- and Spd-2 pathways abolishes mitosis-specific gamma-tubulin recruitment, resulting in low, but not complete loss of gamma-tubulin at centrosomes. The authors show that these low-gamma-tubulin centrosomes are still able to organize microtubules, but these microtubules have different dynamics. Inspired by existing literature in flies and other model organisms, the authors identify Msps/Xmap215 as an important nucleation factor in this scenario.

Major points:

1) The authors use fly embryos with mutant Grip71, Grip75 and Grip163 alleles, which are central to the study. Most conclusions are based on the assumption that some mutants contain only gamma-TuSC, whereas wildtype cells contain a mix of gamma-TuSC and gamma-TuRC. It would be important to show sucrose gradient analyses of extracts to confirm the expected presence/absence of gamma-TuSC/gamma-TuRC.

We agree that it would be nice to perform sucrose gradient analysis of γ -tubulin mutants in different mutant backgrounds, but unfortunately this is not as easy as the Reviewer may think. To clarify, we have used larval brain cells (not embryos) for the analysis of γ -tubulin recruitment to centrosomes. We cannot use embryos because most mutant combinations are lethal beyond larval stages, meaning that mutant adult females are not available for embryo collection (embryos use maternally loaded proteins and mRNA and so it is the genotype of the mother that is important). Performing sucrose gradients with larval brain extracts would be extremely challenging, if not impossible, because a relatively large amount of starting material is required for sucrose gradient centrifugation, and manually dissecting and preparing hundreds if not thousands of larval brains is unrealistic, especially as mutant larvae are rare.

Given that we are not able to carry out these experiments, we have modified the text to include the caveat that some higher-order complexes may partially form in certain mutants. For example, in relation to the ability of Grip71 to recruit γ -TuSCs in *cnn,grip75,grip163* mutants, the text now reads: "Thus, Spd-2 appears to recruit a very small amount of γ -TuSCs (which may, or may not, be present as larger assemblies due to an association with Grip128- γ -tubulin) via Grip71 (i.e. the recruitment that occurs in *cnn,grip75^{GCP4},grip163^{GCP6}* cells), but its recruitment of γ -tubulin complexes relies predominantly on the GCP4/5/4/6 core."

Nevertheless, the most important conclusion is that Cnn can recruit γ -TuSCs independent of pre-formed cytosolic γ -TuRCs and this is based on the finding from one particular mutant – the *spd-2.grip71.grip75.grip128.grip163* mutant – where γ -tubulin levels at mitotic centrosomes are only very slightly reduced compared to wild-type *spd-2* mutants (Figure 1B). This conclusion is based on three assumptions that we argue are all very reasonable:

Assumption 1: flies depleted of 2, if not all 3, GCP4/5/4/6 core components (*grip75.grip128.grip163*) do not have a functioning GCP4/5/4/6 core. The *Grip75^{GCP4}* allele is a null mutant and is combined with a deficiency chromosome that depletes the whole *Grip75^{GCP4}* gene, and the *Grip163^{GCP6}* allele is a very strong depletion allele and is also combined with a deficiency chromosome that depletes the whole *Grip163^{GCP6}* gene. Even if the efficiency of the RNAi against *Grip128^{GCP5}* were poor, it would be hard to form a GCP4/5/4/6 core without *Grip75^{GCP4}* and in the near absence of *Grip163^{GCP6}* (which together provide 3 of the 4 molecules of the complex, including the outermost ones).

Assumption 2: cells depleted of the GCP4/5/4/6 core cannot assemble cytosolic γ -TuRCs. This is reasonable given that even individual depletion of *Grip75^{GCP4}*, *Grip128^{GCP5}* or *Grip163^{GCP6}* already strongly reduces the presence of cytosolic γ -TuRCs (Vogt et al., 2006; Vérollet et al., 2006). In *spd-2.grip71.grip75.grip128.grip163* mutant brain cells, the only γ -TuRC protein not targeted, except for the γ -TuSC components, is Actin (Mozart 1 is expressed only in testes (Tovey et al., 2018) and *Mzt2* does not exist in flies). In *Xenopus* and humans, Actin appears to facilitate γ -TuRC assembly via interactions with a GCP6-N-term-*Mzt1* module, and so it would be unlikely to allow γ -TuSC assembly into higher-order complexes without GCP6 (i.e. *Grip163^{GCP6}*) and *Mzt1*.

Assumption 3: Were Cnn not able to recruit γ -TuSCs independently of pre-formed γ -TuRCs, we would expect a much stronger reduction in γ -tubulin recruitment to centrosomes in *spd-2.grip71.grip75.grip128.grip163* mutant cells. It is reasonable to assume, even without sucrose gradients, that the assembly of γ -TuRCs is strongly impeded in *spd-2.grip71.grip75.grip128.grip163* mutant cells. Nevertheless, γ -tubulin is still recruited to centrosomes at ~66% compared to ~77% in *spd-2* single mutant cells. While statistically significant (as stated in the updated manuscript), this reduction would surely be much greater were Cnn not able to recruit γ -TuSCs.

In the absence of experimental data, we have therefore made these arguments in the main text by making some text modifications and adding a new paragraph, as follows:

*“...the centrosomes in *spd-2.grip71.grip75^{GCP4}.grip128^{GCP5}-RNAi.grip163^{GCP6}* mutant cells had ~66% of the γ -tubulin levels found at wild-type centrosomes, only slightly lower than ~77% in *spd-2* mutants alone (Figure 1A,B). Thus, the recruitment of γ -tubulin to mitotic centrosomes that occurs in the absence of *Spd-2*, i.e. that depends upon Cnn, does not appear to require *Grip71* or the GCP4/5/4/6 core.*

*While we cannot rule out that residual amounts of GCP4/5/4/6 core components in *spd-2.grip71.grip75^{GCP4}.grip128^{GCP5}-RNAi.grip163^{GCP6}* mutant cells may support a certain level of γ -TuSC oligomerisation in the cytosol, we favour the conclusion that Cnn can recruit γ -TuSCs directly to centrosomes in the absence of the GCP4/5/4/6 core for several reasons: First, the alleles used for *grip71* and *grip75^{GCP4}* are null mutants, and the allele for *grip163^{GCP6}* is a severe depletion allele (see Methods), and even individual mutations in, or RNAi-directed depletion of, *Grip75^{GCP4}*, *Grip128^{GCP5}* or *Grip163^{GCP6}* are sufficient to strongly reduce the presence cytosolic γ -TuRCs (Vogt et al., 2006; Vérollet et al., 2006). Second, *spd-2.grip71.grip75^{GCP4}.grip128^{GCP5}-RNAi.grip163^{GCP6}* mutant cells are depleted for all structural γ -TuRC components except for γ -TuSCs and Actin (note that *Mozart1 (Mzt1)* is not expressed in larval brain cells (Tovey et al., 2018) and that *Mzt2* has not been identified in flies). In human and *Xenopus* γ -TuRCs, Actin supports γ -TuRC assembly via interactions with a GCP6-N-term-*Mzt1* module (Liu et al., 2019; Wieczorek et al., 2019, 2020; Zimmermann et al., 2020; Consolati et al., 2020), and so Actin alone is unlikely to facilitate assembly of γ -TuSCs into higher order structures. Third, our data agree with the observation that near complete depletion of *Grip71*, *Grip75^{GCP4}*, *Grip128^{GCP5}*, and *Grip163^{GCP6}* from S2 cells does not prevent γ -tubulin recruitment to centrosomes (Vérollet et al., 2006). Fourth, given the strength of mutant alleles used, one would have expected a much larger decrease in centrosomal γ -tubulin levels in *spd-2.grip71.grip75^{GCP4}.grip128^{GCP5}-RNAi.grip163^{GCP6}* mutant cells were Cnn not able to recruit γ -TuSCs directly to centrosomes. Thus, Cnn appears to recruit γ -TuSCs to centrosomes without a requirement for them to first assemble into higher-order complexes.”*

2) Given the advantage of the *Cnn Δ CM1* separation of function mutant, I do not understand why it is not used throughout the study. Instead, full Cnn loss is used, which results in strongly reduced *Spd-2* levels (Figure 2C,D). Are the observed differences between wild-type and mutants in Figure 2-5 dependent on defective PCM or do they also occur in a *Cnn Δ CM1* background?

We generated stocks to analyse γ -tubulin recruitment and microtubule dynamics in *cnn ^{Δ CM1}.grip71.grip163* cells. As predicted, we found that there was no accumulation of γ -tubulin at mitotic centrosomes in *cnn ^{Δ CM1}.grip71.grip163* mutant cells (new Figure 4A,B). We therefore proceeded with CherryTemp experiments with *cnn ^{Δ CM1}.grip71.grip163* mutant cells expressing GFP-PACT (to mark centrosomes) and Jupiter-mCherry (to mark microtubules). The data is shown in new Figure 4C-G. As you can see, the dynamics of the microtubules organised by centrosomes in *cnn ^{Δ CM1}.grip71.grip163* cells are similar to *cnn.grip71.grip163* cells, with some interesting differences. Unlike in *cnn.grip71.grip163* mutant cells, the centrosomal Jup-mCherry signal prior to cooling is slightly higher than in controls, indicating that these mutant centrosomes organise microtubules robustly. Once again, the microtubules depolymerise much slower in *cnn ^{Δ CM1}.grip71.grip163* mutant cells compared to controls but, somewhat unexpectedly, it was even harder to depolymerise the microtubules completely. We had concluded before that microtubules were more cold-resistant when nucleated independently

of γ -TuRCs and this new data further supports this important point, as well as showing that the presence of Cnn, albeit Cnn ^{Δ CM1}, makes the microtubules even more cold-resistant. Interestingly, microtubule re-growth at centrosomes in *cnn^{\Delta}CM1, grip71, grip163* cells is still ~3 times slower compared to controls, indicating that the presence of Cnn ^{Δ CM1} does not help γ -TuRC-independent microtubule nucleation/polymerisation. It may be that the CM1 domain, in addition to its role in γ -TuRC recruitment, plays a role in recruiting other factors that facilitate γ -TuRC-independent microtubule nucleation. Indeed, Tim Megraw's paper from 2009 showed that depleting the CM1 domain affected the centrosomal recruitment of TACC and Msps (Zhang et al., 2009). Moreover, deleting the CM1 domain appears to affect the ability of Cnn to form a proper PCM scaffold (see the response to Reviewer 3 point 2 below). Thus, it is hard to have a definitive answer for whether the observed differences between control and *cnn, grip71, grip163* mutant cells are dependent on defective PCM or not, as the PCM is likely also defective in *cnn^{\Delta}CM1, grip71, grip163* mutants. Nevertheless, we think this new data is very useful as it supports our data in the original submission and helps make the important point that microtubules nucleated by different mechanisms have different dynamic properties.

3) Statistical tests should support the conclusions in the text. If the authors claim differences between different genetic backgrounds (e.g. that *spd2*-mutants only have ~77% of gamma-tubulin at mitotic centrosomes compared to wild-type), statistical tests must compare mutant mitosis vs. wild-type mitosis.

We agree. We have now carried out the appropriate statistical tests and included them in the new version of the paper. For more detail, see the response to Reviewer 2 point 2.

4) While Cnn, *grip71*, *grip163* mutants do not accumulate gamma-tubulin at centrosomes in mitosis, they still have low levels of centrosomal gamma-tubulin. It is therefore misleading to refer to "gamma-tubulin negative centrosomes".

This is a fair point. While we suspect this small fraction of γ -tubulin is non-functional in regard to microtubule nucleation i.e. it is the interphase pool of γ -tubulin and interphase centrosomes do not organise microtubules, we agree that referring to them as "gamma-tubulin negative centrosomes" is misleading. We have now changed the text to refer to them simply as "*cnn, grip71, grip163* mutant centrosomes" or "*cnn, grip71, grip163* centrosomes".

Minor points:

1) The abstract states that gamma-TuRC is a catalyst of microtubule nucleation. By definition, a catalyst takes part in a reaction but is not part of the final product. Although our knowledge of the nucleation mechanism is still incomplete, mechanistic studies suggest a non-catalytical mechanism since gamma-TuRC was found to stay attached to the microtubule end after nucleation (Consolati et al. 2020, Wieczorek et al. 2020).

We have now removed any reference to the γ -TuRC being a catalyst.

2) Cnn Δ CM1 flies: genotyping data should be provided besides describing gRNAs.

We are not entirely sure what the Reviewer means here. We had already stated in the main text and methods that the deletion region spanned from R98 to D167. For further clarity, we now included the word "inclusive" in both the main text and the methods: main text: "*We therefore deleted the CM1 domain (amino acids 98-167, inclusive) from the endogenous cnn gene (see Methods) from the endogenous cnn gene...*"; Methods: "*R98 to D167, inclusive*". Please do let us know if further information is required.

3) Is it important to combine *spd-2* with all four mutants, *grip75* *grip128* *grip163* and *grip71*? What about *spd-2* *grip71* cells and *spd-2* *grip75* *grip128* *grip163* cells? Should that not have the same effect?

This comment relates to Major point 1, as our main conclusion (that Cnn can recruit γ -TuSCs) is only possible when combining *spd2* with all four mutants i.e. targeting all γ -TuRC specific proteins is the most likely way to deplete as many pre-formed γ -TuRCs as possible. Depleting only *Spd-2* and *Grip71* would leave fully assembled γ -TuRCs in the cytosol, as assembly does not require *Grip71*. Depleting *Spd-2*, *Grip75*, *Grip128*, and *Grip163* would prevent cytosolic γ -TuRC assembly, but there is a possibility that *Grip71* may still act as a link between γ -TuSCs and Cnn. It was therefore necessary to deplete *Spd-2*, *Grip75*, *Grip128*, and *Grip163*, and *Grip71*.

4) CM1-containing factors are the only known factors able to directly bind and activate gamma-TuRC. How do the authors envision activation of gamma-TuRC in the absence of Cnn?

This is a good question but remains unanswered. Phosphorylation of γ -TuRCs is the most obvious possibility. For example, Aurora A phosphorylates NEDD1 (homologue of *Grip71*) to promote microtubule nucleation (Pinyol et al., 2013). NME7 kinase has been shown to increase the activity of purified γ -TuRCs (Liu et al., 2014). Other γ -TuRC components are also phosphorylated, but the consequences on γ -TuRC activity are not known. Another possibility is that TOG proteins indirectly promote the closing of the γ -TuRCs while adding tubulin dimers onto γ -tubulin (Thawani et al., 2020).

5) Do the authors think that each identified pathway to microtubule nucleation (i.e. *Spd-2/gamma-TuRC*, Cnn/ γ -TuSC, Msps/*mei38*) as revealed by mutant genetic backgrounds contributes to a similar extent to overall nucleation capacity also in an unperturbed genetic background?

Another good question, but it is very difficult to answer. Our view is that when γ -TuRCs are present and active they will likely dominate microtubule nucleation, out-competing the ability of TOG domain proteins to stimulate

microtubule nucleation independently of γ -TuRCs. Nevertheless, TOG proteins will likely help promote microtubule nucleation from γ -TuRCs when both are present, as has been previously shown *in vitro* (Thawani et al., 2018; King et al., 2020; Consolati et al., 2020) and in fission yeast (Flor-Parra et al., 2018). We also believe that both Spd-2 and Cnn γ -TuRC recruitment pathways will contribute simultaneously. Another question is whether Cnn recruits γ -TuRCs instead of γ -TuSCs when γ -TuRCs are present in the cytosol. We assume this will depend on Cnn's affinity for γ -TuRCs versus γ -TuSCs and on the relative levels of γ -TuRCs and γ -TuSCs in the cytosol.

6) How does CM1 mediate binding to gamma-TuRC? Using recombinant Cnn fragments, the authors find that a Cnn triple mutant (R101Q, E102A and F115A) no longer binds gamma-tubulin, suggesting these residues together mediate binding to gamma-tubulin complexes. However, it is not tested to what extent R101, E102 and F115 individually contribute to gamma-tubulin binding. Does the binding mode in *Drosophila* resemble more the one in humans or in budding yeast? Also, was this done with extracts from Grip71, Grip75, Grip128RNAi, Grip163 embryos or normal embryos?

We apologise for not stating that the IPs were carried out using wild-type embryos extracts – we have now included this information in the main text, methods and figure legend. We have now tested the relative contributions of R101, E102 and F115 and show the results in Figure 1G. We find that mutating only F115 abolishes binding, while mutating both R101 and E102 together reduces but does not abolish binding. Thus, binding of fly CM1 to γ -TuRCs appears to be similar to that of human CM1.

7) Figure 2C: Should the green channel not correspond to Spd-2?
Thank you for pointing out this mistake – now corrected.

8) I suggest to reconsider the color-coding of graphs. While the colored background of the dot plots in Figure 1 and 2 are a matter of taste, the coloring of graphs in Figure 4F-H is confusing. Here, genetic backgrounds of fly lines are colored in the same way as the microscopy channels in Figure 4A-E, but they do not belong together. We have now modified the colour-coding of images/graphs in the Figure, as suggested – note that this is now Figure 5A-E.

9) A *tacc* mutant allele is used in experiments, but is not further described. Please provide the necessary background information.

We thank the reviewer for pointing this out. We had also forgotten to include the *mmps* alleles used. The information for *mmps* and *tacc* are now included in the methods.

10) The authors assess spindle quality in Cnn, grip71, grip163 cells and show that spindle quality worsens with ectopic *mmps*. For comparison it would be good to compare spindle quality side by side with a wild-type situation. This data is now included in Figure S5A,B.

11) Introduction: "[...], however, as they depend upon each other for their proper localisation within the PCM and act redundantly." - Sentence is incomplete.

I think this was just to do with how we had phrased the sentence (the position of "however" was confusing). We have now rephrased the sentence: "*It is difficult to determine the individual role of these proteins in γ -TuRC recruitment, as they act redundantly and depend on each other for their proper localisation within the PCM*".

12) Introduction: "Cnn contains the highly conserved CM1 domain (Sawin et al., 2004), which binds directly to γ -tubulin complexes in yeast and humans (Brilot et al., 2021; Wieczorek et al., 2019)". - Choi et al 2010 should also be cited here.

This citation has been added.

13) Results: "Typically, interphase centrosomes have only ~5-20% of the γ -tubulin levels found at mitotic centrosomes, [...]". - Citation is needed

We now cite our Conduit et al., 2014 paper.

14) The authors should discuss that *Mmps* was found to act non-redundantly with gamma-tubulin in interphase nucleation (Rogers, MBC, 2008), contrary to the conclusions in the current manuscript.

Thank you for pointing this out. We have now modified the relevant part of the discussion to read:

"TOG domain and TPX2 proteins have been shown to work together with γ -TuRCs (or microtubule seed templates) to promote microtubule nucleation (Thawani et al., 2018; Flor-Parra et al., 2018; Gunzelmann et al., 2018b; Consolati et al., 2020; King et al., 2020; Wieczorek et al., 2015). Consistent with this, co-depletion of γ -tubulin and the *Drosophila* TOG domain protein *Mmps* did not delay non-centrosomal microtubule regrowth after cooling compared to single depletions in interphase S2 cells (Rogers et al., 2008). Nevertheless, several studies, mainly *in vitro*, have shown that TOG and TPX2 proteins can also function independently of γ -TuRCs to promote microtubule nucleation (Roostalu et al., 2015; Woodruff et al., 2017; Schatz et al., 2003; Slep and Vale, 2007; Ghosh et al., 2013; Thawani et al., 2018; King et al., 2020; Zheng et al., 2020; Tsuchiya and Goshima, 2021). Our data suggest that, unlike from non-centrosomal sites in interphase S2 cells, *Mmps* can promote γ -TuRC-independent microtubule nucleation from centrosomes in

mitotic larval brain cells. This difference may reflect Msps having a high local concentration at centrosomes.

Referees cross-commenting

This is a good paper in my opinion, they need to add some controls though, to determine the expected presence/absence of gTuSC/gTuRC in the different mutants. An important advance is the finding that gTuSC can function as nucleator in parallel to gTuRC, depending on the recruitment mechanism. Different recruitment mechanisms, nucleation templates, and regulatory strategies co-exist and provide complex regulation and robustness to nucleation/spindle assembly.

We thank the Reviewer for their thorough and constructive review. We hope they will agree to allow publication without us having to perform the sucrose gradient experiments that, as discussed above, will be very difficult, if not impossible, to carry out.

Reviewer #1 (Significance (Required)):

This is a very well-executed study and the data is presented clearly. However, some findings would benefit from additional experiments to substantiate the main interpretations. If these points are addressed, the study would provide an important conceptual advance in the field, namely that animal cells may rely on two different gamma-tubulin complexes for nucleation at mitotic centrosomes, gamma-TuSC and gamma-TuRC, which differ not only in their composition of GCP proteins but also the mode of recruitment to the centrosome. The findings will be of interest to all cell biologists.

Reviewer #2 (Evidence, reproducibility and clarity (Required)):

Summary

This paper sets out to further our understanding of how two proteins, Cnn and Spd-2, independently recruit g-Tubulin ring complexes(g-TuRC) to mitotic centrosomes in *Drosophila* cells. It uses some robust classical genetics to generate mutants to reduce/remove GCP4/5/6, Dgrip71 and Cnn and Spd-2 from cells, monitoring the consequences using live imaging.

It begins by showing that Cnn can recruit g-Tubulin independently of the core g-TuRC components or Dgrip71, and that a mutant Cnn lacking the CM1 domain cannot, strongly suggesting that, similarly to other organisms, the CM1 domain is essential for this function.

It then demonstrates that Spd-2, in contrast, cannot localise g-Tubulin in the absence of the g-TuRC components or Dgrip71.

In the second half of the paper, then use this tool as a proxy for centrosomes that completely lack mitotic g-Tubulin recruitment, in order to explore spindle assembly in the absence of centrosomal g-Tubulin. The show that microtubules and spindle are still nucleated but do so with different dynamics. This section is particularly convincing, given the use of the live de/repolymerisation assays using the CherryTemp device.

Finally, the authors visualise spindle formation in the absence of centrosomal g-Tubulin, alongside a number of other MT associated proteins, including Msps.

Major Comments

1. The claims and conclusions relating to the first half of the paper are supported by the data, but they need to be caveated by a clear explanation of the alleles used. Some are well-characterised mutant lines but have they been previously shown to completely remove the associated protein products? For the RNAi lines, do the authors have evidence (via Western blots) that these remove the protein products? It is not necessary that they show Western blots for all the lines, and it does not invalidate the major conclusions that the fly line carrying mutations in *cnn*, *grip71*, *grip163* completely fails to localise g-Tubulin to mitotic centrosomes. However, they need to help the reader understand much more clearly whether these lines are complete nulls and, consequently this may impact the strength of their interpretation of the relationship between Grip163 versus Grip75, discussed both at the end of the relevant section and in the Discussion.

We appreciate the reviewer's concern and have now included a detailed description in the Methods section of the alleles we use and their known effect on protein levels (pasted below for convenience). We have also included western blots for *cnn* and *spd2* mutants to show the absence of detectable protein in larval brains. Unfortunately, we could not provide western blots for the other mutants, as we don't have working antibodies for these proteins (although for Grip71 we did make an antibody and did western blots that showed the absence of protein in *grip71* mutants, but this antibody has now been commercialised and so the western blot is published on the CRB website: <https://crbdiscovery.com/polyclonal-antibodies/anti-grip71-antibody/>). Nevertheless, protein levels for the *grip75*, *grip163*, *msps* and *tacc* mutants have been shown previously (now cited in the new text). We have also

modified the main text to allow the reader to better understand whether proteins are completely absent or strongly reduced. In response to the specific comment about interpreting the relationship between Grip163 and Grip75, as we mention in the new methods section, the Grip75 allele is a null mutant while the Grip163 mutant is a severe depletion; thus, the fact that the Grip163 mutant has a stronger effect on γ -tubulin recruitment is not due to a stronger depletion.

New text in methods: “For *spd-2* mutants, we used the *dspd-2*^{Z35711} mutant allele, which carries an early stop codon resulting in a predicted 56aa protein. Homozygous *dspd-2*^{Z35711} mutant flies lack detectable Spd-2 protein on western blots and so the allele is therefore considered to be a null mutant (Giansanti et al., 2008). This allele no longer produces homozygous flies (which is common for mutant alleles kept as balanced stocks for many years), which combined *dspd-2*^{Z35711} with a deficiency that includes the entire *spd-2* gene (*dspd-2*^{Df(3L)st-j7}). On western blots, there was no detectable Spd-2 protein in extracts from *dspd-2*^{Z35711} / *dspd-2*^{Df(3L)st-j7} hemizygous mutant brains (Figure S4B). For *cnn* mutants, we combined the *cnn*^{f04547} and *cnn*^{HK21} mutant alleles. The *cnn*^{f04547} allele carries a piggyBac insertion in the middle of the *cnn* gene and is predicted to disrupt long *Cnn* isoforms, including the centrosomal isoform (*Cnn-C* or *Cnn-PA*) (Lucas and Raff, 2007). This mutation is considered to be a null mutant for the long *Cnn* isoforms (Lucas and Raff, 2007; Conduit et al., 2014). The *cnn*^{HK21} allele carries an early stop codon after *Cnn-C*'s Q78 (Vaizel-Ohayon and Schejter, 1999) and affects both long and short *Cnn* isoforms – it is considered to be a null mutant (Eisman et al., 2009; Chen et al., 2017a). On western blots, there was no detectable *Cnn-C* protein in *cnn*^{f04547} / *cnn*^{HK21} hemizygous mutant brains (Figure S4A). For Grip71, we used the *grip71*¹²⁰ mutant allele, which is a result of an imprecise p-element excision event that led to the removal of the entire *grip71* coding sequence except for the last 12bp; it is considered to be a null mutant (Reschen et al., 2012). We combined this with an allele carrying a deficiency that includes the entire *grip71* gene (*grip71*^{Df(2L)Exel6041}). On western blots, there is no detectable Grip71 protein in *grip71*¹²⁰ / *grip71*^{Df(2L)Exel6041} hemizygous mutant brains (see blots on CRB website, which were performed by us). For Grip75^{GCP4}, we used the *grip75*¹⁷⁵ mutant allele, which carries an early stop codon after Q291. Homozygous *grip75*¹⁷⁵ mutant flies lack detectable Grip75^{GCP4} protein on western blots and so the allele is therefore considered to be a null mutant (Schnorrer et al., 2002). We combined this with an allele carrying a deficiency that includes the entire *grip75*^{GCP4} gene (*grip75*^{Df(2L)Exel7048}). In the absence of a working antibody, we have not confirmed the expected absence of Grip75^{GCP4} protein in *grip75*¹⁷⁵ / *grip75*^{Df(2L)Exel7048} hemizygous mutant flies on western blots. For Grip128^{GCP5}, we used the UAS-controlled *grip128*-RNAi^{V29074} RNAi line, which is part of the VDRC's GD collection, and drove its expression using the *Insc-Gal4* driver (BL8751), which is expressed in larval neuroblasts and their progeny. In the absence of a working antibody, we have not confirmed the absence or reduction of Grip128^{GCP5} protein on western blots. RNAi was used for *grip128*^{GCP5} as its position on the X chromosome made generating stocks with multiple alleles technically challenging. For Grip163^{GCP6}, we used the *grip163*^{GE2708} mutant allele, which carries a p-element insertion between amino acids 822 and 823 (total protein length is 1351aa) and behaves as a null or strong hypomorph mutant (Vérollet et al., 2006). We combined this with an allele carrying a deficiency that includes the entire *grip163*^{GCP6} gene (*grip163*^{Df(5L)Exel6115}). In the absence of a working antibody, we have not confirmed the absence or reduction of Grip163^{GCP6} protein in *grip163*^{GE2708} / *grip163*^{Df(5L)Exel6115} hemizygous mutant flies on western blots. For *Msp*s, we used the *mshps*^p and *mshps*^{MJ15} mutant alleles. The *mshps*^p allele carries a p-element insertion within, or close to, the 5' UTR of the *mshps* gene and results in a strong reduction, but not elimination, of *Msp*s protein on western blots (Cullen et al., 1999). The *mshps*^{MJ15} allele was generated by re-mobilisation of the p-element (the genetic consequence of which has not been defined) and also results in a strong reduction, but not elimination, of *Msp*s protein on western blots (Cullen et al., 1999; Lee et al., 2001). For TACC, we used the *tacc*^{stella} allele which contain a p-element insertion of unknown localisation but that results in no detectable TACC protein on western blots (Barros et al., 2005). For Mei-38, we used the UAS-controlled *mei-38*-RNAi^{JHM23752} RNAi line, which is part of the NIG's TRIP Valium 20 collection, and drove its expression using the *Insc-Gal4* driver (BL8751). In the absence of a working antibody, we have not confirmed the absence or reduction of Mei-38 protein on western blots. RNAi was used for *mei-38* as its position on the X chromosome made generating stocks with multiple alleles technically challenging. Moreover, the only available mutant of *mei-38* affects a neighbouring gene.”

2. I have an issue with the statistics in Figure 1 & 2. I realise the t-tests in Figure 1 show the significant differences between g-Tubulin recruitment to centrosomes in interphase and mitosis, in order to demonstrate the difference between the Spd-2;Grip combination line in (B) and the Spd-2; CnnCM1 double mutant in (D). But in doing so, it draws attention to the fact that there is no similar t-test between mitotic g-Tubulin recruitment to centrosomes in WT, Spd-2 and the Spd-2;Grip combination lines. This lack of stats between conditions is further confused by the language used in the text: In the Figure legend, the authors claim mitotic centrosomal g-Tubulin levels between are WT, Spd-2 and the Spd-2;Grip combination lines "similar", and in the text they say: the *spd-2* Grip combination line had g-Tubulin "similar to the levels found at *spd-2* mutants alone". But then they give numbers - an average of 77% of wild type for *spd2* and 66% of wild type for the *spd-2* Grip combination. I'm sure if they did a t-test they would find a significant difference between these conditions. This doesn't invalidate the thrust of what they're claiming, but they do need to be consistent in language, analysis and interpretation. We agree that we should have performed a statistical comparison between the γ -tubulin levels for "WT mitosis" vs "*spd2* mitosis" and for "*spd-2* mitosis" vs "*spd2.grip71.grip75.grip128.grip163* mitosis" (Figure 1B). We have now done this and found statistically significant differences in both cases. We have included the new p-values in the figure and modified the main text to read: "In fact, the centrosomes in these *spd-*

2.grip71,grip75^{GCP4},grip128^{GCP5}-RNAi,grip163^{GCP6} mutant cells had ~66% of the γ -tubulin levels found at wild-type centrosomes, only slightly lower than the levels found at spd-2 mutants alone (Figure 1A,B)."; and we have modified the legend to read: "A one-way ANOVA with a Sidak's multiple comparisons test was used to make the comparisons indicated by p values in the graph. Note that there is only a small reduction in mitotic centrosomal γ -tubulin levels in spd-2 mutants and in spd-2, grip71,grip75^{GCP4}, grip128^{GCP5}-RNAi,grip163^{GCP6} mutants, showing that Cnn can still efficiently recruit γ -tubulin complexes to mitotic centrosomes when only γ -TuSCs are present." Note that due to performing comparisons multiple times with the same data sets, it was necessary to use a one-way ANOVA with a Sidak's multiple comparisons test (rather than paired t-tests).

For Figure 1D, we did not compare WT mitosis vs cnn^{ACM1} spd-2 mitosis, as the point here was to test whether there was an increase from interphase to mitosis in cnn^{ACM1} spd-2 mutants and we wanted to maintain the statistical power of using a paired t-test (one is more likely to detect differences with a paired t-test than with a multiple comparisons ANOVA, making the conclusion that there is no difference between interphase and mitotic cnn^{ACM1} spd-2 centrosomes even more solid).

Similarly, in Figure 2, it would be better to assess any statistically significant difference between mitotic accumulation of g-Tubulin between fly lines, rather than accumulation between interphase and mitosis (which is pretty clear cut). This would help to clarify whether differences between loss of grip subunits are merely additive or synergistic. Again, this doesn't invalidate the overall result that concomitant loss of cnn, grip71 and grip163 completely abolishes mitotic centrosomal accumulation of g-Tubulin, but it is a more complete analysis of the extant data.

As for Figure 2, we respectfully disagree that we should make comparisons between genotypes instead of, or in addition to, making comparisons between interphase and mitotic centrosomes within the same genotype. This is because we will lose statistical power by performing a multiple comparisons test. Indeed, if we were to compare both within and between selected genotypes (14 comparisons in total), then we lose the statistically significant differences between interphase and mitotic centrosomes in cnn,grip75,grip163 (p=0.04) and cnn,grip71,grip75 (p=0.08) genotypes, when there clearly appears to be a difference (as stated by the Reviewer). Given that the point of this experiment is to elucidate which proteins are required to allow maturation from interphase to mitosis, rather than which combination of mutations has the stronger effect, we feel that maintaining the paired t-test analysis is more appropriate.

3. One OPTIONAL experiment that would significantly improve the study would be similar CherryTemp live imaging of the cells lacking both centrosomal g-Tubulin and Msps. Currently the manuscript finishes with a fixed analysis of MT de/repolymerisation in these cells, which provides evidence that Msps has a role in MT nucleation in the absence of centrosomal g-Tubulin-nucleated MTs, but very little else can be concluded.

We would love to do this experiment but the genetics are complicated. We would have to generate stocks containing a cnn,grip71,GFP-PACT triple allele chromosome II and a grip163,msps,Jupiter-mCherry triple allele chromosome III. While live data would provide interesting insights into the dynamics of microtubules nucleated in the absence of γ -TuRCs and reduction of Msps, our fixed analysis is at least sufficient to implicate Msps in γ -TuRC-independent microtubule organisation. Moreover, the cnn, grip71,grip163 mutations impede centrosome assembly to some degree. Thus, in future our aim is to identify the way in which Spd-2 recruits γ -TuRCs so that we can eliminate this pathway with more subtle mutations. Ideally, we could then combine the CnnF115A mutant (that we have now identified as being sufficient to eliminate Cnn binding to γ -TuRCs) with a Spd-2 mutant that also can't bind γ -TuRCs. We could then combine these mutations with msps mutants and perform the live experiments suggested by the Reviewer. I'm sure you will agree, however, that we are still a long way from this.

4. There is, perhaps surprisingly, no mention of Augmin in the paper. Augmin has been shown to recruit g-TuRC to pre-existing MTs, through the grip71 subunit (Chen et al., 2017). So, presumably, in cnn, grip71, grip163, g-Tubulin cannot be recruited to pre-existing MTs either? This could add impact to the results - in that it implies the MT nucleation seen in the absence of cnn, grip71 and grip163 actually reflects, not just loss of centrosome function, but also loss of Augmin function. Mentioning this in the discussion could help increase the impact of the paper.

We apologise for this oversight. Indeed, it is perfectly possible that Grip71/Augmin-mediated amplification of microtubules during microtubule re-growth from centrosomes could influence the difference in recovery rates between control and mutant centrosomes. We have now include the following in the results section: "Moreover, we note that the absence of Grip71 may impact the ability of Augmin to amplify the microtubules being nucleate from centrosomes, thereby reducing nucleation efficiency compared to controls". We also add later when assessing spindle assembly differences between cnn,grip71,grip163 and cnn,grip71,grip163,msps mutants: "Note also that the absence of Grip71 abolishes the Augmin-mediated nucleation pathway necessary for efficient spindle assembly (Reschen et al., 2012; Chen et al., 2017b; Dobbelaere et al., 2008; Vérollet et al., 2006), but as both mutant types lacked Grip71 this cannot explain the differences observed between the mutants."

Minor comments

1. The cnn, grip71, grip163 mutant image in Fig3 B after 40 min cooling appears to have 4 centrioles. Is this a cell that exited and re-entered mitosis?

Cnn mutant cells often have centrosome segregation problems, resulting in cells with variable numbers of centrioles (Conduit et al., 2010b, Current Biology). We have now mentioned this in the legends for Figure 3, Figure 5, and Figure S5.

2. Methods should contain more detail on the de/repolymerisation live imaging analysis (including the numbers of cells contributing to the analysis) and techniques such exponential curve fitting.

We have now included this information in the methods and updated this information in the figure legend (to include cell numbers, not just centrosome numbers, and to indicate that GraphPad Prism was used to generate the models.

3. P5 para 2 - "GPC4/5/4/6" should read "GCP4/5/6"

We actually use the GCP4/5/4/6 nomenclature throughout as it represents the 2 copies of GCP4 to one copy of GCP5 and GCP6 in the complex, as well as the order of these molecules.

4. Fig legend 1 - "error bar" should read "scale bar"

Thanks, now corrected

Reviewer #2 (Significance (Required)):

The experimental approach (genetics and cell biology) taken in this manuscript is very appropriate and the experiments are of high quality. It uses the strengths of Drosophila to cleverly engineer flies to pull apart the relationship between two different ways to recruit the main MT nucleator, g-Tubulin, to mitotic centrosomes. This is an important advance for the specific research field of centrosome biology.

By generating a fly that completely fails to localise g-Tubulin to mitotic centrosomes, the paper is able to explore whether MTs and the mitotic spindle can form in its absence. Again, there is very high quality imaging and image analysis, using a commercially available (but very cool) fast heating/cooling apparatus - the CherryTemp to explore the dynamics of MT generation. The limitation to this approach, though, is that g-Tubulin itself is still present and presumably able to nucleate MTs in the cytosol or elsewhere, albeit inefficiently. As such, it adds to a body of centrosomal and cell division research, rather than adding a highly significant conceptual advance.

Similarly, the finding that Msps is involved in nucleating MTs in the absence of centrosomal g-Tubulin, via fixed analysis, supports other work, rather than moving the field forwards.

Overall, assuming the caveats mentioned in the major comments are dealt with, I see this as a robust and very well carried out piece of research, that will be of interest to those investigating the broad field of cell division

My field of expertise is Drosophila cell division

We thank the Reviewer for their thorough and constructive review. We hope that the reviewer may agree with us and the other Reviewers that revealing the complexity of γ -TuRC recruitment and microtubule nucleation at centrosomes, particularly the findings that different types of γ -tubulin complexes are recruited to centrosomes by different tethering proteins and that microtubules nucleated by different mechanisms have different dynamic properties, provide important conceptual advances.

Reviewer #3 (Evidence, reproducibility and clarity (Required)):

Centrosomes are complex and it has been appreciated for some time that they likely nucleate microtubules by more than one mechanism. However, what these mechanisms exactly are, and which are the most significant has not been clear. A major contributor to centrosomal microtubule nucleation the tubulin isoform gamma-tubulin (g-tubulin), which is present in two complexes, a smaller gTuSC that contains gamma tubulin along with GCP2 and 3 and a larger g-tubulin ring complex (gTuRC) whose assembly additionally requires GCP4/5/6. A second high-level question has been whether the centrosome has any g-tubulin-independent microtubule nucleation mechanisms. In this manuscript, the authors use a collection of mutants and RNAi conditions in the Drosophila brain to generate a picture of centrosomal microtubule nucleation pathways. They show that there are two g-tubulin-dependent and a third g-tubulin-independent microtubule nucleation pathways. They show that the first g-tubulin-dependent pathway depends on the CM1 domain of the centrosomal PCM matrix protein, Centrosomin (Cnn) and on the gTuSC components GCP2/3, but not on the components specifically required for gTuRC assembly. The second g-tubulin-dependent pathway depends on Spd-2 (and not Cnn) and requires the gTuRC-specific components and NEDD1/Grip71. By inhibiting both of these pathways, the authors also show that there is a robust g-tubulin-independent microtubule nucleation pathway. Overall, has the potential to be an impactful contribution from a conceptual point-of-view. I would be excited to recommend publication if the major comments below, particularly points 1 and 2, could be addressed.

1. The experiment in Fig. 2B examines what is required for Spd-2 to recruit g-tubulin to mitotic centrosomes that

lack Cnn. This panel should include a *cnn* mutant-only control, for which the readers are currently referred to an older paper from 2014. Without repeating this control in parallel to one of the conditions in this panel, it is impossible to say whether the addition of the *grip71* mutation has any effect on γ -tubulin levels.

We have now included a *cnn* mutant-only control in Figure 2A,B. We performed a new experiment comparing γ -tubulin recruitment in *cnn* mutant vs *cnn,grip71* mutant cells (samples processed together with brains paired on the same slide). The mitotic levels of γ -tubulin at *cnn* mutant centrosomes were on average ~1.5 fold higher than those at *cnn,grip71* mutant centrosomes and this difference was statistically significant ($p=0.0396$, paired t-test). This is actually a very similar difference to what I had previously observed as a postdoc, but I had not included the data in my Conduit et al., 2014 eLife paper. Thus, I am now very confident that the addition of the *grip71* mutation does indeed have an effect on γ -tubulin levels, albeit a relatively mild effect.

2. The experiment in Fig. 2B is in the background of a Cnn loss-of-function mutation in which centrosomal Spd-2 is at just under 40% of its levels in brains with Cnn (according to Fig. 2D). So the Spd-2 doing the recruiting is the non-Cnn-dependent population. The authors should also do one experiment in the background of their Cnn-CM1delete mutant or their Cnn CM1 γ -tubulin recruitment mutant, because these backgrounds would be expected to have normal amounts of Cnn matrix and normal levels of Spd-2. Comparing the amount of γ -tubulin recruitment in a *cnn* loss-of-function mutant to that in a *cnn-CM1delete* mutant would reveal whether the Cnn-bound Spd-2 can contribute to γ -tubulin recruitment in the same way that the Cnn-independent Spd-2 can. These two populations could easily differ in their ability to recruit γ -tubulin. Also, is it clear that these two pathways can act in parallel (i.e. that assembly of the Cnn matrix around the centriole does not mask the ability of Cnn-independent Spd-2 to recruit γ -tubulin)? Thus, there are three possibilities- all interesting- for the outcome of this experiment. The Cnn-CM1delete mutant/Cnn-CM1 γ -tubulin recruitment mutants could: (1) recruit less γ -tubulin than the *cnn* loss-of-function mutant (if Cnn matrix assembly inhibits the Cnn-independent Spd-2 pathway), (2) recruit the same amount of γ -tubulin as the *cnn* loss-of-function mutant (if the Cnn matrix does not inhibit the Cnn-independent Spd-2 pathway but Cnn-dependent Spd-2 does not itself recruit γ -tubulin), or (3) recruit more γ -tubulin than the *cnn* loss-of-function mutant (if both the Cnn-dependent and Cnn-independent Spd-2 can recruit γ -tubulin).

We have now compared γ -tubulin levels at centrosomes in *cnn* null and *cnn*^{ΔCM1} mutant cells to test the ability of the Cnn-dependent population of Spd-2 to recruit γ -tubulin. We found that there was a small but statistically significant increase in the centrosomal levels of γ -tubulin in *cnn*^{ΔCM1} mutant cells compared to *cnn* null mutant cells (Figure S1A,B). Thus, the Cnn-dependent population of centrosomal Spd-2 does seem to recruit at least some γ -tubulin complexes. We suspect that the difference was not greater because the PCM in *cnn*^{ΔCM1} mutant cells appears to be perturbed (as we see an offset between the γ -tubulin signal and the Asl signal in these cells (Figure S1A)). This offset is indicative of a destabilised PCM (Lucas et al., 2007) and so suggests that Cnn^{ΔCM1} cannot form a robust centrosome scaffold. This was somewhat unexpected, but not unreasonable given the large deletion. We report these findings in the first paragraph of the Spd-2 recruitment section.

3. The paper needs a summary model figure that the field can understand. The current model in Fig. 2E does not suffice in this regard. It would be nice to have this model appear at the end of the paper to outline the 3 pathways for centrosomal microtubule nucleation outlined by the work. Maybe have an arc for the centrosome at the bottom of the figure and show arrows from the gTuSC to the Cnn CM1 domain from the gTuRC to the Cnn CM1 domain and the gTuRC to Spd-2 or something like this. How you draw this could be impacted by the experiment outlined above in point 2. Also, there would be a γ -tubulin-independent pathway in the figure. Not everyone reads papers carefully, and you want people to be able to get the takeaway message at a glance.

We have now completely modified the Figure and moved the model to the end of the paper (new Figure 6). We thank the Reviewer for this suggestion as it really does provide a clearer message for the reader.

4. The authors show that this pathway is modulated by loss of Minispindles (MSPs)-but as this is a critical microtubule assembly factor, it seems likely that MSPs loss might modulate all of the pathways. From the data in Figure 4, my main takeaway would be that MSPs is not the central player in the γ -tubulin independent nucleation pathway. It might make the paper more impactful to end the story after Fig. 4, move the current Fig. 5 to the supplement and add a nice model figure at the end.

We agree that MSPs may play a role beyond microtubule nucleation, including plus end growth, and that this may also influence the efficiency of spindle formation in *cnn,grip71,grip163,mmps* mutants. Nevertheless, our microtubule regrowth data in original Figure 5A (now Figure 5I) clearly show that MSPs is a key player in the γ -tubulin independent nucleation pathway at centrosomes. Perhaps the Reviewer missed this point as the data was in Figure 5 and not the original Figure 4. Moreover, the original Figure 5E (now Figure S5C) shows that the effect of depleting MSPs in addition to *cnn, grip71* and *grip163* is specific to cells containing centrosomes i.e. if MSPs played a significant role in microtubule regulation beyond its role at centrosomes, then one would expect spindle formation to be worse when comparing mutant cells that lack centrosomes.

Given the potential for misunderstanding, we have now grouped together the data on centrosome microtubule organisation at steady state and the data on microtubule regrowth from centrosomes within *cnn,grip71,grip163* vs *cnn,grip71,grip163,mmps* mutants – these data are now together in the new Figure 5. We have also moved the

spindle assembly data from original Figure 5C-E to a new supplementary Figure (Figure S5). We then end the paper on a model figure in new Figure 6.

Minor comments:

5. In Fig. 1E the sequence labels are confusing. Please label each sequence on the left with the residue numbers in the corresponding endogenous protein that are shown in the alignment.

You are absolutely right, I'm not sure why our labelling was like that. Now corrected.

6. In Fig. 1F, please label with location of molecular weight markers

Now added.

Reviewer #3 (Significance (Required)):

Repeating my text from above. Centrosomes are complex and it has been appreciated for some time that they likely nucleate microtubules by more than one mechanism. However, what these mechanisms exactly are, and which are the most significant has not been clear. A major contributor to centrosomal microtubule nucleation the tubulin isoform gamma-tubulin (g-tubulin), which is present in two complexes, a smaller gTuSC that contains gamma tubulin along with GCP2 and 3 and a larger g-tubulin ring complex (gTuRC) whose assembly additionally requires GCP4/5/6. A second high-level question has been whether the centrosome has any g-tubulin-independent microtubule nucleation mechanisms. In this manuscript, the authors use a collection of mutants and RNAi conditions in the Drosophila brain to generate a picture of centrosomal microtubule nucleation pathways. They show that there are two g-tubulin-dependent and a third g-tubulin-independent microtubule nucleation pathways. They show that the first g-tubulin-dependent pathway depends on the CM1 domain of the centrosomal PCM matrix protein, Centrosomin (Cnn) and on the gTuSC components GCP2/3, but not on the components specifically required for gTuRC assembly. The second g-tubulin-dependent pathway depends on Spd-2 (and not Cnn) and requires the gTuRC-specific components and NEDD1/Grip71. By inhibiting both of these pathways, the authors also show that there is a robust g-tubulin-independent microtubule nucleation pathway. Overall, has the potential to be an impactful contribution from a conceptual point-of-view. I would be excited to recommend publication if the major comments, particularly points 1 and 2, could be addressed.

I hope you will agree that we have now adequately addressed all concerns and that the work is ready for publication in JCB.

We want to thank you once again for taking the time to carefully consider the manuscript and provide constructive criticism that has clearly improved the manuscript.

With kind regards

Paul Conduit

June 28, 2023

Re: JCB manuscript #202212043R

Dr. Paul T Conduit
Institut Jacques Monod
15 rue Hélène Brion
Paris 75205
France

Dear Paul,

Thank you for submitting your revised manuscript entitled "Multifaceted modes of γ -tubulin complex recruitment and microtubule nucleation at mitotic centrosomes" and thank you for your patience. The manuscript has been seen by two of the original reviewers whose full comments are appended below. Please note that reviewer #2 had agreed to re-review the paper but was delayed in returning their report so, in the interests of expediting the decision as much as we can (and given that this reviewer was generally supportive of the paper from the first round), we decided to move forward with just the two reports.

In any case, the remaining reviewers are both supportive of the work. Prior to reaching a final decision, we would like you to address the remaining point from Rev 1 verifying the Cnn Δ mutant genotype and demonstrating that this mutant is expressed comparably to wild type Cnn. We do not expect any further analysis with this mutant, simply verification that it was indeed generated and expressed.

Our general policy is that papers are considered through only one revision cycle; however, given that the suggested changes are relatively minor we are open to one additional short round of revision. Please note that I will expect to make a final decision without additional reviewer input upon resubmission.

Please submit the final revision within one month, along with a cover letter that includes a point by point response to the remaining reviewer comments.

Thank you for this interesting contribution to Journal of Cell Biology. You can contact me or the scientific editor listed below at the journal office with any questions, cellbio@rockefeller.edu.

Sincerely,

Arshad Desai, PhD
Monitoring Editor
Journal of Cell Biology

Tim Spencer, PhD
Executive Editor
Journal of Cell Biology

Reviewer #1 (Comments to the Authors (Required)):

The authors have addressed most of my concerns and here I am only referring to the remaining issues. Major point 2 and minor point 2 (both are related):

My previous comment and authors' reply:

2) Given the advantage of the Cnn Δ CM1 separation of function mutant, I do not understand why it is not used throughout the study. Instead, full Cnn loss is used, which results in strongly reduced Spd-2 levels (Figure 2C,D). Are the observed differences between wild-type and mutants in Figure 2-5 dependent on defective PCM or do they also occur in a Cnn Δ CM1 background?

We generated stocks to analyse γ -tubulin recruitment and microtubule dynamics in *cnn Δ CM1,grip71,grip163* cells. As predicted, we found that there was no accumulation of γ -tubulin at mitotic centrosomes in *cnn Δ CM1,grip71,grip163* mutant cells (new Figure 4A,B). We therefore proceeded with CherryTemp experiments with *cnn Δ CM1,grip71,grip163* mutant cells expressing GFP-PACT (to mark centrosomes) and Jupiter-mCherry (to mark microtubules). The data is shown in new Figure 4C-G. As you

can see, the dynamics of the microtubules organised by centrosomes in *cnnΔCM1,grip71,grip163* cells are similar to *cnn,grip71,grip163* cells, with some interesting differences. Unlike in *cnn,grip71,grip163* mutant cells, the centrosomal Jup-mCherry signal prior to cooling is slightly higher than in controls, indicating that these mutant centrosomes organise microtubules robustly. Once again, the microtubules depolymerise much slower in *cnnΔCM1,grip71,grip163* mutant cells compared to controls but, somewhat unexpectedly, it was even harder to depolymerise the microtubules completely. We had concluded before that microtubules were more cold-resistant when nucleated independently of γ -TuRCs and this new data further supports this important point, as well as showing that the presence of Cnn, albeit *CnnΔCM1*, makes the microtubules even more cold-resistant. Interestingly, microtubule re-growth at centrosomes in *cnnΔCM1,grip71,grip163* cells is still ~3 times slower compared to controls, indicating that the presence of *CnnΔCM1* does not help γ -TuRC-independent microtubule nucleation/polymerisation. It may be that the CM1 domain, in addition to its role in γ -TuRC recruitment, plays a role in recruiting other factors that facilitate γ -TuRC-independent microtubule nucleation. Indeed, Tim Megraw's paper from 2009 showed that depleting the CM1 domain affected the centrosomal recruitment of TACC and Msps (Zhang et al., 2009). Moreover, deleting the CM1 domain appears to affect the ability of Cnn to form a proper PCM scaffold (see the response to Reviewer 3 point 2 below). Thus, it is hard to have a definitive answer for whether the observed differences between control and *cnn,grip71,grip163* mutant cells are dependent on defective PCM or not, as the PCM is likely also defective in *cnnΔCM1,grip71,grip163* mutants. Nevertheless, we think this new data is very useful as it supports our data in the original submission and helps make the important point that microtubules nucleated by different mechanisms have different dynamic properties.

My new comment:

Independently of these considerations, it is currently not shown that the *Cnn* CM1 deletion mutant was correctly generated and is expressed properly. The whole point of the mutant was to separate two major *Cnn* functions (PCM scaffold vs gTuSC/gTuRC binding; see also point 2 of reviewer 3), but without the supporting evidence (see also minor point 2 below) it is currently unclear whether this has been achieved. These data are important for supporting the major conclusion that CM1 (as opposed to the entire *Cnn* protein and therefore potentially CM1-unrelated mechanisms) is essential for g-tubulin mitotic centrosome recruitment in the absence of the Spd-2-dependent pathway.

Previous comment and authors' reply:

2) *CnnΔCM1* flies: genotyping data should be provided besides describing gRNAs.

We are not entirely sure what the Reviewer means here. We had already stated in the main text and methods that the deletion region spanned from R98 to D167. For further clarity, we now included the word "inclusive" in both the main text and the methods: main text: "We therefore deleted the CM1 domain (amino acids 98-167, inclusive) from the endogenous *cnn* gene (see Methods) from the endogenous *cnn* gene..."; Methods: "R98 to D167, inclusive". Please do let us know if further information is required.

My new comment:

This mutant is the result of genome editing and the manuscript should include the experimental evidence that the CM1 encoding region is deleted (e.g. PCR, sequencing,...) and that the mutant *Cnn* protein is expressed at levels similar to the wild type (ideally by western). In my opinion, this should be standard for validating newly generated mutants. In addition, this validation is also crucial for correct interpretation of the results. For example, the mutant protein may not be expressed or may be expressed at a lower level, in which case the phenotypes may be related to a general loss of *Cnn* function and not a specific loss of binding of CM1 to gTuRC. Validation of mutant protein expression is particularly important considering that the authors seem to believe that there may be a reduction in the PCM scaffold also with the CM1 deletion mutant (which in my opinion is surprising, although no direct evidence is shown).

Reviewer #3 (Comments to the Authors (Required)):

All of my concerns have been addressed. The manuscript will make an excellent contribution to the JCB.

Minor Comments:

The *Drosophila* nomenclature is tough going for non-*Drosophila* people that work on centrosomes. More consistent superscripting of the *Drosophila* names with the homolog names (like *grip71* with NEDD1 or *Mei-38* with TPX2) would make it easier to read.

There is a typo on line 74-Should be "Spc110" not "Spd110"

July 18, 2023

RE: JCB Manuscript #202212043RR

Dr. Paul T Conduit
Institut Jacques Monod
15 rue Hélène Brion
Paris 75205
France

Dear Paul:

Thank you for submitting your revised manuscript entitled "Multifaceted modes of γ -tubulin complex recruitment and microtubule nucleation at mitotic centrosome" and thank you for your patience. We have now assessed your revised paper and we would be happy to publish your paper in JCB pending final revisions necessary to meet our formatting guidelines (see details below).

A. MANUSCRIPT ORGANIZATION AND FORMATTING:

1) Text limits: Character count for Articles is < 40,000, not including spaces. Count includes the abstract, introduction, results, discussion, and acknowledgments. Count does not include title page, materials and methods, figure legends, references, tables, or supplemental legends. Your paper is slightly below this limit at the moment but please bear it in mind when revising.

2) Figure formatting: Scale bars must be present on all microscopy images, including inset magnifications. Molecular weight or nucleic acid size markers must be included on all gel electrophoresis. Please add molecular weight markers to the blot in supplementary figure 6.

3) Statistical analysis: Error bars on graphic representations of numerical data must be clearly described in the figure legend. The number of independent data points (n) represented in a graph must be indicated in the legend. Statistical methods should be explained in full in the materials and methods. For figures presenting pooled data the statistical measure should be defined in the figure legends. Please also be sure to indicate the statistical tests used in each of your experiments (both in the figure legend itself and in a separate methods section) as well as the parameters of the test (for example, if you ran a t-test, please indicate if it was one- or two-sided, etc.).

****Also, since you used parametric tests in your study (e.g. t-tests, ANOVA, etc.), you should have first determined whether the data was normally distributed before selecting that test. In the stats section of the methods, please indicate how you tested for normality. If you did not test for normality, you must state something to the effect that "Data distribution was assumed to be normal but this was not formally tested."****

4) Materials and methods: Should be comprehensive and not simply reference a previous publication for details on how an experiment was performed. Please provide full descriptions (at least in brief) in the text for readers who may not have access to referenced manuscripts. The text should not refer to methods "...as previously described."

5) Please be sure to provide the sequences for all of your primers/oligos and RNAi constructs in the materials and methods. You must also indicate in the methods the source, species, and catalog numbers (where appropriate) for all of your antibodies.

6) Microscope image acquisition: The following information must be provided about the acquisition and processing of images:

- a. Make and model of microscope
- b. Type, magnification, and numerical aperture of the objective lenses
- c. Temperature
- d. imaging medium
- e. Fluorochromes
- f. Camera make and model
- g. Acquisition software
- h. Any software used for image processing subsequent to data acquisition. Please include details and types of operations involved (e.g., type of deconvolution, 3D reconstitutions, surface or volume rendering, gamma adjustments, etc.).

7) References: There is no limit to the number of references cited in a manuscript. References should be cited parenthetically in

the text by author and year of publication. Abbreviate the names of journals according to PubMed.

8) Supplemental materials: There are normally strict limits on the allowable amount of supplemental data. Articles may usually have up to 5 supplemental figures. However, given the current circumstances, we will allow you to have the extra supplementary figure. However, please do not add to the current number.

****Please also note that tables, like figures, should be provided as individual, editable files. A summary of all supplemental material (that is, in addition to the supplementary figure legends) should appear at the end of the Materials and methods section. Please see any recent JCB paper for an example of this.****

9) eTOC summary: A ~40-50 word summary that describes the context and significance of the findings for a general readership should be included on the title page. The statement should be written in the present tense and refer to the work in the third person. It should begin with "First author name(s) et al..." to match our preferred style.

10) Conflict of interest statement: JCB requires inclusion of a statement in the acknowledgements regarding competing financial interests. If no competing financial interests exist, please include the following statement: "The authors declare no competing financial interests." If competing interests are declared, please follow your statement of these competing interests with the following statement: "The authors declare no further competing financial interests."

11) A separate author contribution section is required following the Acknowledgments in all research manuscripts. All authors should be mentioned and designated by their first and middle initials and full surnames. We encourage use of the CRediT nomenclature (<https://casrai.org/credit/>).

12) ORCID IDs: ORCID IDs are unique identifiers allowing researchers to create a record of their various scholarly contributions in a single place. At resubmission of your final files, please consider providing an ORCID ID for as many contributing authors as possible.

13) Please note that JCB now requires authors to submit Source Data used to generate figures containing gels and Western blots with all revised manuscripts. This Source Data consists of fully uncropped and unprocessed images for each gel/blot displayed in the main and supplemental figures. Since your paper includes cropped gel and/or blot images, please be sure to provide one Source Data file for each figure that contains gels and/or blots along with your revised manuscript files. File names for Source Data figures should be alphanumeric without any spaces or special characters (i.e., SourceDataF#, where F# refers to the associated main figure number or SourceDataFS# for those associated with Supplementary figures). The lanes of the gels/blots should be labeled as they are in the associated figure, the place where cropping was applied should be marked (with a box), and molecular weight/size standards should be labeled wherever possible. Source Data files will be made available to reviewers during evaluation of revised manuscripts and, if your paper is eventually published in JCB, the files will be directly linked to specific figures in the published article.

14) Journal of Cell Biology now requires a data availability statement for all research article submissions. These statements will be published in the article directly above the Acknowledgments. The statement should address all data underlying the research presented in the manuscript. Please visit the JCB instructions for authors for guidelines and examples of statements at (<https://rupress.org/jcb/pages/editorial-policies#data-availability-statement>).

B. FINAL FILES:

****It is JCB policy that if requested, original data images must be made available to the editors. Failure to provide original images upon request will result in unavoidable delays in publication. Please ensure that you have access to all original data images prior to final submission.****

****The license to publish form must be signed before your manuscript can be sent to production. A link to the electronic license to publish form will be sent to the corresponding author only. Please take a moment to check your funder requirements before choosing the appropriate license.****

Please contact the journal office with any questions, cellbio@rockefeller.edu.

Thank you for this interesting contribution, we look forward to publishing your paper in Journal of Cell Biology.

Sincerely,

Arshad Desai, PhD
Monitoring Editor
Journal of Cell Biology

Tim Spencer, PhD
Executive Editor
Journal of Cell Biology